# Quantitative assessment of changes in surface particulate matter concentrations and precursor emissions over China during the COVID-19 pandemic and their implications for Chinese economic activity

Hyun Cheol Kim[1,2,*], Soontae Kim[3], Mark Cohen[1], Changhan Bae[4], Dasom Lee[5], Rick Saylor[1], Minah Bae[3], Eunhye Kim[3], Byeong-Uk Kim[6], Jin-Ho Yoon[5], and Ariel Stein[1]

[1] Air Resources Laboratory, National Oceanic and Atmospheric Administration, College Park, MD, USA
[2] Cooperative Institute for Satellite Earth System Studies, University of Maryland, College Park, MD, USA
[3] Department of Environmental and Safety Engineering, Ajou University, Suwon, South Korea
[4] National Air Emission Inventory and Research Center, Sejong, South Korea
[5] School of Earth Sciences and Environmental Engineering, Gwangju Institute of Science and Technology, Gwangju, South Korea
[6] Georgia Environmental Protection Division, Atlanta, GA, USA

*Correspondence to*: Hyun Cheol Kim (hyun.kim@noaa.gov)

**Abstract.** Sixty days after the lockdown of Hubei province, where the coronavirus was first reported, China's true recovery from the pandemic remained an outstanding question. This study investigates how human activity changed during this period using observations of surface pollutants. By combining surface data with a three-dimensional chemistry model, the impacts of meteorological variations and variations in yearly emissions control are minimized, demonstrating how pollutant levels over China changed before and after the Lunar New Year from 2017 to 2020. The results show that the reduction in $NO_2$ concentrations, an indicator of emissions in the transportation sector, was clearly deeper and longer in 2020 than in normal years, and started to recover after February 15. By contrast, $PM_{2.5}$ emissions had not yet recovered by the end of March, showing a reduction around 30% compared with normal years. $SO_2$ emissions had not affected significantly by the pandemic. Additional model study using a top-down emissions adjustment still confirms a reduction around 25% in unknown surface $PM_{2.5}$ emissions over the same period, even after realistically updating $SO_2$ and $NO_x$ emissions. This evidence suggests that different economic sectors in China may be recovering at different rates, with the fastest recovery in transportation and a slower recovery likely in agriculture. The apparent difference between the recovery timelines of $NO_2$ and $PM_{2.5}$ implies that monitoring a single pollutant alone (e.g. $NO_x$ emissions) is insufficient to draw conclusions on the overall recovery of the Chinese economy.

## 1. Introduction

Measuring pollutants can provide empirical and immediate information on human activity compared with traditional survey-based measures, although interpreting spatial and temporal trends in such data is complex. The novel coronavirus SARS-CoV-2 has struck globally since it was first reported in December 2019 in China, the first country to be affected. After strong efforts by the Chinese government, including the lockdown of Hubei province, the outbreak seems to have eased as of the end of March 2020. New daily infections in Hubei have significantly reduced, with reported new cases dropping to zero from the thousands of new cases reported daily in February (Worldometer, 2020), and lockdown restrictions have been eased. As countries around the world struggle to slow outbreaks of the pandemic disease, it becomes important to observe and analyze signals of recovery in economic and public activity in China.

A large proportion of the surface pollutants in China originate from anthropogenic emissions by five major economic sectors: transportation, industry, power generation, residential (cooking and heating), and agriculture (Li et al., 2017). Emissions changes

for different economic sectors can be approximately inferred based on changes in ambient concentrations of specific pollutants if uncertainties associated with real-world emissions and meteorological variations can be reduced or accounted for. $NO_2$ concentration is strongly associated with nitrogen oxide ($NO_x = NO + NO_2$) emissions (Beirle et al., 2011; Georgoulias et al., 2019), and, since mobile sources (transportation) account for a large proportion of $NO_x$ emissions, $NO_2$ concentrations can offer a good proxy for traffic in urban areas (Li et al., 2017). Meanwhile, $SO_2$ emissions are strongly related to the industrial and residential sectors. The agricultural sector plays a critical role in tropospheric chemistry, providing most of the ammonia emissions that contribute to the formation of inorganic aerosols (Pinder et al., 2007).

Surface observations of pollutants provide an independent dataset that can be compared with socioeconomic data based on surveys. Three main components affect variations in pollutant concentrations: (1) natural variations (e.g. short-term synoptic weather, interannual meteorological variations, and long-term climate change) , (2) long-term trends due to emissions control, and (3) sporadic socioeconomic events (Kim et al., 2017a). The coronavirus offers a case of an emissions change caused by an unprecedented, isolated social event. Therefore, signals from these first two components—meteorological variations and year-on-year emissions controls—must be minimized to isolate the true signal of the impact of the pandemic on air pollutant concentrations. A state-of-the-art three-dimensional atmospheric chemistry model can help to separate these confounding factors. This study attempts to estimate the impact of the pandemic on Chinese regional air quality, thus inferring changes in social activity based on observations of surface pollutants.

Although early studies have reported Chinese air quality during the period in question, in terms of surface observations and air quality indices (Bao and Zhang, 2020; Chauhan and Singh, 2020; He et al., 2020; Shi and Brasseur, 2020; Xu et al., 2020), satellite observations (Liu et al., 2020a, 2020b), atmospheric chemistry modeling (Kang et al., 2020; Li et al., 2020; Wang et al., 2020), emissions estimation via inverse modeling (Miyazaki et al., 2020; Zhang et al., 2020), secondary aerosol formation (Huang et al., 2020), and human activity and energy use (Wang and Su, 2020), it remains challenging to fully isolate the impact of the pandemic on the region's air quality. To quantitatively assess changes in major surface pollutants and their precursor emissions over China during the pandemic period, we conducted a series of analyses using surface observations and atmospheric chemistry models, with simulations based on a bottom-up emissions inventory and top-down assimilated emissions. Section 2 describes the observational data recorded from surface monitors and satellite, as well as the baseline modeling methodology. Section 3 describes the methodology for processing time-series data, estimating top-down emissions, and assessing sectoral impacts of emissions. Section 4 presents and discusses the results. Finally, Section 5 summarizes the findings and their implications.

## 2.  Data

### 2.1 Observations

Surface observation data were obtained from the China National Environmental Monitoring Center (CNEMC; data available at http://www.pm25.in). Hourly ambient air concentration data for $PM_{10}$, $PM_{2.5}$, CO, $NO_2$, $O_3$, and $SO_2$ were available for 1,571 sites (over China) and 1,459 sites (within the study domain; **Figure 1**). After removing sites with less than 80% data availability for each year (2017-2020, $\pm$ 60 days of Lunar New Year (LNY)), the analysis used observations from 1,332 sites. Data processing procedures are explained in Section 3.1 and further discussed in Section 4.4.

### 2.2 Satellite

The TROPOspheric Monitoring Instrument (TROPOMI) $NO_2$ vertical column-density, level-2 data (S5P_L2_NO2) were obtained from NASA GES DISC (http://tropomi.gesdisc.eosdis.nasa.gov). TROPOMI is a hyperspectral spectrometer onboard the

75 Sentinel-5P satellite, with wavelength coverage over ultraviolet to visible (270 nm to 495 nm), near infrared (675–775 nm), and shortwave infrared (2305–2385 nm) wavelengths (Eskes et al., 2019; van Geffen et al., 2019). High-quality pixels from level-2 data (3.5×7 km resolution at the nadir) were selected using the quality flags provided by the product (qa_value > 0.75) and then spatially regridded into the study domain using a conservative spatial-regridding method that preserves mass during interpolation (Kim et al., 2018, 2016, 2020).

### 2.3 Model

80 Meteorological and atmospheric chemistry transport models were used over East Asia with 27-km horizontal resolution. The Weather Research and Forecasting Model (WRF, version 3.4.1) was used for meteorological simulations (Skamarock and Klemp, 2008). The National Oceanic and Atmospheric Administration (NOAA) National Centers for Environmental Protection (NCEP) Final Analysis (FNL) product (NCEP, 2000) provided the initial and boundary conditions for the WRF simulations. For chemistry 85 simulations, CMAQ (version 4.7.1) (Byun and Schere, 2006), the Meteorology–Chemistry Interface Processor (MCIP, version 3.6) (Otte and Pleim, 2010), and the Sparse Matrix Operator Kernel Emission (SMOKE) modeling framework were used, employing the meteorological inputs provided by the WRF simulations. **Table 1** details the modeling configurations, and **Figure S1** compares models with observations. The models provide a reasonably realistic simulation of atmospheric chemical and physical processes over the considered domain, especially in terms of their daily variations from 2017 to 2019 (e.g., R = 0.91~0.94 for 90 $PM_{2.5}$, see Emery et al. (2017) for general model performance guidance). However, in 2020, as the effects of the pandemic began to take hold, the chemical model's predictions—based on typical (as opposed to pandemic-influenced) emissions—systematically overpredict pollutant concentrations, consistent with a pandemic-influenced reduction in emissions.

### 2.4 Emissions inventory

This study used two sets of emissions inventories, the Comprehensive Regional Emissions Inventory for Atmospheric Transport 95 Experiment (CREATE, version 2.3) (Jang et al., 2019) and the Model Inter-Comparison Study for Asia (MICS-Asia) emissions inventory (MIX inventory, 2010) (Li et al., 2017). While the CREATE inventory is based on the latest information, including the 2016 KORUS-AQ campaign (https://espo.nasa.gov/korus-aq), the MIX inventory has been tested in many diverse applications (Li et al., 2019a, 2019b; Zhang et al., 2017). The time series analysis in **Figure 2** (discussed below) was based on the CREATE emissions inventory, but the results of the analysis did not depend in any significant way on a choice between these two base 100 inventories. The CREATE inventory is provided as an annual mean for each Chinese province for the year 2016 and the SMOKE preprocessor was used to convert the inventory to hourly model-ready inputs. The base model simulations use these 2016 emissions for the entire 2017-2020 modeling period.

### 3. Method

This section describes the following aspects of the analysis: (1) data-processing procedures for analyzing the time series, (2) 105 emissions-adjustment procedures to update $SO_2$ and $NO_x$ emissions to near real-time, and (3) brute-force modeling procedures to estimate Chinese emissions by sector. It should be noted that the time series analysis (discussed in Section 4.1) utilizes fixed emissions inventory (i.e. bottom-up emissions inventory) and the emission adjustment experiment (Section 4.2) utilizes observation-based top-down emissions. Sectoral emissions estimations method is for Section 4.3.

### 3.1 Time-series analysis

Four types of variation (meteorological, weekly, yearly, and the Chinese spring festival) were reduced or accounted for in the surface observations, as follows. Meteorological influences were reduced by combining surface data with output from a three-dimensional chemistry model to calculate estimated emissions. Since the model simulations with fixed emissions inventory respond to the variations of meteorological conditions, we can infer the relationship between emissions and ambient pollutant concentrations under a specific weather condition. By applying this relationship, we convert the changes of observed concentrations

into the changes of emissions. Weekly variations, a unique feature of anthropogenic emissions, were removed by using a seven-day moving average. The impact of the Chinese spring festival, the biggest traditional holiday celebrating Lunar New Year (LNY), was normalized by rearranging the time series to center on the LNY in each solar year. The LNY alignment was necessary to account for the irregular happening of the LNY dates. Seven-day moving average filtering was also required to avoid unfair comparisons between different weekdays after the LNY alignment. Otherwise, we may compare different weekdays for different

year (e.g. 2020 LNY on January 25, Saturday and 2019 LNY is February 5, Tuesday). **Figure S4** shows that the seven-day moving average filter smooths but does not significantly change the time-series results. Finally, yearly emission variations were removed by setting a base period (-60 to -10 days before LNY) and calculating relative changes from the average of the base period.

We followed the data-processing procedures suggested by Bae et al. (2020) for their emissions-updating system (hereafter BAE2020). First, the observational and modeled data were paired and tested, and observation sites with more than 20% of values

missing were discarded. To avoid over-weighting dense urban sites, observations occurring within the same model grid cell were averaged. Second, weekly variations were removed using seven-day moving averages, and the impact of the Chinese spring festival was normalized by rearranging the time series to center on LNY in each year. Third, meteorological variations were removed by applying the ratio between observed and modeled concentrations. Using a simple linear assumption, observed pollutant concentrations were combined with the results of the chemical model to create estimates of actual emissions that are less sensitive

to meteorological variations. Use of the linear assumption in the concentration-to-emission conversion is further discussed in Section 4.4 The total estimated emissions, $E_{est}$, and their relative variations, $rE_{est}$, were calculated as:

$$E_{est}(t) = E_{mod} \cdot \frac{C_{obs}(t)}{C_{mod}(t)} \tag{1}$$

$$rE_{est}(t) = \frac{E_{est}(t)}{\sum_{t=base} E_{est}(t)/n_{base}} \times 100\% \tag{2}$$

where $C$ is the daily pollutant concentration; $t$ is days from LNY; $base$ and $n_{base}$ are the pre-LNY base period (shown in **Figure**

**2**) and its number of days, respectively; and $E_{mod}$ is the model emissions. To normalize the yearly changes, a base period (-60 to -10 days before LNY) was set, with relative changes calculated from the average of that base period (i.e., $rE_{est}(t)$). The impact of the pandemic was inferred by calculating the difference in estimated emissions between normal years and 2020. Since the model uses a fixed emissions inventory for each year, $E_{mod}$ cancels out in the comparison.

For the spatial analyses of the data (e.g. **Figure 3**), points data were converted to area format. Similar to the time series data

processing, the observational and modeled data were paired and tested. Considering location of each paired data set, we assigned point data to their corresponding Chinese prefecture. By averaging all concentrations in each prefecture, we constructed the prefecture-level concentration data set (for each prefecture polygon), which were then converted into domain grids using a conservative spatial-regridding technique. Section 4.4 further discusses the data-processing procedures.

**3.2 Top-down emissions adjustment**

For the second analyses (discussed in Section 4.2), we updated major pollutant emissions to more realistic level and analysed simulated chemical behaviors. Due to stringent emissions control policies by the Chinese government, Chinese anthropogenic emissions changed dramatically over recent years. For example, the annual mean surface $SO_2$ concentration across China was 8.4 ppb in 2016, but dropped less than half of this level (3.7 ppb) in 2020. To incorporate a realistic change in emissions from 2016 to 2020, we applied observation-based emissions adjustment factors to the 2016 CREATE emissions inventory to reproduce

emissions in 2020. In general, model emissions can be adjusted based on the ratios between observed and modeled surface concentrations:

$$\frac{E_{adj}}{E_{mod}} = \beta \cdot \frac{C_{obs}}{C_{mod}} \tag{3}$$

where $\beta$ is a sensitivity factor in the emission-to-concentration conversion. $\beta$ is close to 1 if less secondary chemical reactions are

involved. BAE2020 assumed a fixed $\beta = 1$ to update $SO_2$ emissions, and they demonstrated that the adjusted emissions effectively reproduced surface $SO_2$ concentrations over China. Similar approaches were also confirmed to be effective for the $NO_x$ emissions adjustment over the same East Asian domain using satellite-based measurements of $NO_2$ column densities (Bae et al., 2020a; Chang et al., 2016).

While this simple assumption works practically, we tried to conduct the emission adjustment processing more carefully,

considering the unprecedent changes of chemical environment during the pandemic period. We extend the approach of BAE2020, offering two major enhancements. First, we calculate daily emissions-adjustment factors to represent the rapid changes in emissions under the pandemic situation. We applied 14-day moving averages to avoid uncertainties caused by insufficient data points day to day. Second, we calculated spatial and temporal variations in $\beta$ and then applied these to the emissions-adjustment factors. **Table 2** compares the data-processing steps used in this study with those used in BAE2020.

The $\beta$ values are calculated as follows. In the real world, the sensitivity of concentration to changes in emissions is not unique or spatially homogeneous (i.e., $\beta \neq 1$), especially for $NO_x$ emissions and $NO_2$ concentrations. $\beta$ values for specific location and time can be calculated if we have two model simulations with different emissions applied. Previous studies have calculated $\beta$ values for a model by using changes in concentration caused by a certain amount of perturbed emissions (e.g., Lamsal et al., 2011 used a 15% emissions pertubation).

To obtain more realistic $\beta$ values, we have conducted two model simulations, *base* and *adj1* runs. First, the *base* model simulation was conducted using normal emissions inventory, CREATE, we have introduced previously. The second simulation, *adj1* run, was conducted using perturbed emissions to estimate how the model responds according to the change of emissions. We adjusted emissions according to the ratio between observed and modelled surface concentrations, so we can reproduce more realistic chemical environment.

From these two simulations, the *base* and *adj1* runs, we calculate the emissions-to-concentration sensitivity, $\beta$ values, in specific spatial and temporal scale – for each Chinses prefecture daily. $\beta$ values are calculated as,

$$\beta_{p,t} = \frac{[E_{adj1}/E_{base}]_{p,t}}{[C_{adj1}/C_{base}]_{p,t}} \tag{4}$$

where $p$ and $t$ stand for indices of Chinese prefectures and specific dates. Using calculated $\beta$ values for each prefecture and date, we finally obtain the adjusted emissions for the second and final simulations, *adj2* run.

$$[E_{adj2}]_{p,t} = \beta_{p,t} \cdot \left[ \frac{C_{obs}}{C_{base}} \cdot E_{base} \right]_{p,t} \tag{5}$$

We further discuss the characteristics of the emissions-to-concentration sensitivity in Section 4.4.2.

**3.3 Estimation of sectoral contributions**

The contributions of emissions from each sector to surface PM$_{2.5}$ concentrations over China were estimated using the brute-force method (BFM), an approach that uses changes in modeled outputs as a result of perturbed emission inputs (Burr and Zhang, 2011). The MIX emissions inventory provides information on five sectors: residential, industry, power generation, transportation, and agriculture. Sectoral contributions were calculated by applying the perturbed emissions for each sector:

$$\text{Contr. (sector)} = \frac{(C_{base} - C_{\Delta E, sector})/\Delta E}{C_{base}} \times 100\ \% \tag{7}$$

where C is the surface PM2.5 concentration and $\Delta E$ is the ratio of the emissions perturbations. A 50% reduction was chosen to perturb emissions for each individual sector. Fractional contributions of each emission sector were calculated compared to the sum of all five emissions sector contributions. Application of the BFM to East Asian air quality models and a discussion of its uncertainties has been presented elsewhere (Kim et al., 2017b).

## 4. Results

**4.1 Time-series analysis**

Reducing meteorological, weekly, and yearly variations, as well as variations resulting from the Chinese spring festival made the comparison of pandemic-influenced surface observations to normal conditions more robust and useful. Estimated NO$_x$ emissions (**Figure 2**) display variations from the spring festival season. From 2017–2019, the estimated NO$_x$ emissions demonstrate a clear reduction during the festival period (by up to 45% between -10 and +20 days from LNY). In 2020, this reduction is slightly deeper and continues longer, implying that the coronavirus outbreak further reduced traffic in China. The difference between the estimated emissions in the 2017–2019 time series and those in the 2020 time series in **Figure 2** reflects the relative significance of the impact of the coronavirus (p < 0.01 for t-test of comparison after LNY).

Interestingly, the 2020 time series (that is, the combined effect of the spring festival and the coronavirus) remains flat from the LNY to February 15. As both effects likely overlapped, they appear inseparable during the period. The maximum impact from the coronavirus seen in the data is a 58% reduction on February 15, 2020 from that seen in prior, baseline years (2017–2019). The level of NO$_x$ emissions from February 1 to 15 (close to a 50% reduction) might suggest a floor level for reduced emissions under current conditions in terms of technology and infrastructure. This might have important implications for chemical modeling and emissions control, perhaps implying a floor for emissions reductions that China can realistically reach under current conditions. The blue line represents a time series from Hubei only (46 sites), showing, as would be expected, that the impact in Hubei has been more significant and sustained.

The reduced NO$_x$ emissions began to increase after February 15, almost recovering to their normal level by the end of March 2020. Hence, the impact of the coronavirus pandemic on NO$_x$ emissions in China lasted almost two months. **Figure 3** shows the spatial distribution of the estimated changes in NO$_x$ emissions from the base period to the period of maximum impact, January 25–February 14, 2020, and the recovery period, February 24–March 15, 2020. Just after LNY, NO$_x$ emissions strongly reduce across China, but their inferred recovery is spatially inhomogeneous. As shown in **Figure 3**, Hubei province continued to show a strong reduction (by more than 50%) compared with the pre-LNY level, even in the recovery phase (period 2). Other regions show various patterns in NO$_x$ levels compared with previous years. These observations are consistent with space-borne, remote-sensing measurements from the TROPOMI (**Figure S1**). Similar to the surface observations in **Figure 3**, the spatial distributions of NO$_2$ column densities during the period of maximum impact (January 25 – February 14) and the recovery period (February 24 – March 15) were generated as changes from the baseline period (November 26, 2019 – January 15, 2020).

The impact of the virus may actually have begun before the spring festival. In normal years (2017–2019), variation in estimated pre-LNY baseline period (-60 to -10 days) NO$_x$ emissions is relatively small, because the model uses fixed emissions and weekly variations have already been removed. However, the estimated emissions in 2020 are relatively low starting from about 15 days before LNY, and this relative reduction is more pronounced in Hubei. This suggests that our baseline period in 2020 already includes a partial coronavirus impact. If this is true, the impact of the pandemic would be even stronger than inferred here, as it is based on a year-by-year comparison of concentrations during and after the typical-year base-period.

Unlike the temporal trend in NO$_x$ emissions and their spatial distribution, comparison of changes in the PM$_{2.5}$ level suggests a different story (**Figure 2b**). Contrary to NO$_x$ emissions, PM$_{2.5}$ concentrations typically show a slight increase near LNY, likely due to increased PM$_{2.5}$ emissions from fireworks, a long-held tradition in China (Kong et al., 2015), and show only a relatively moderate reduction from typical levels (by 10–20%) over the remainder of the spring festival. Unlike NO$_x$ emissions, the case of PM$_{2.5}$ involves both direct emissions of particulate matter and gas-to-particle conversion of emitted precursors (e.g., SO$_2$, NO$_x$, NH$_3$, VOCs) mediated by atmospheric chemical transformations. As discussed in the Methods section, we assume the same approximate relationship for PM$_{2.5}$ as with NO$_x$ between the ambient observations and their associated emissions. This approach suggests that emissions decreased by roughly 30% from normal levels through the end of March to reach $72.7 \pm 6.6\%$ of the 2017–2019 level from February 4 to March 25, 2020. Interestingly, the pandemic does not seem to have significantly affected SO$_2$ emissions (see **Figure S3**), suggesting that the pandemic's effects on the power generation and industrial sectors have been relatively small.

**4.2 Experiment with updated SO$_2$ & NO$_x$ emissions**

As discussed in Section 3.2 above, we used an alternative approach to investigate unidentified PM$_{2.5}$ emissions, specifically applying more realistic SO$_2$ and NO$_x$ emissions adjustments. Using this methodology, we repeated CMAQ simulations with SO$_2$ and NO$_x$ emissions adjusted based on surface measurements. Daily and prefecture-level emission-adjustment factors were calculated and applied to the baseline emissions inventory. The two CMAQ simulations—a baseline simulation with the CREATE emissions inventory and an adjustment simulation with updated emissions—were both compared with observations from surface-monitoring sites (**Figure 4**). Individual site comparisons are also available in **Figure S11**.

For both SO$_2$ and NO$_2$ concentrations, the CMAQ simulation with adjusted emissions performed well, reproducing observed variations in surface concentrations. It should be noted that the CREATE v2.3 emissions inventory we used was constructed for 2016 and applied to a 2020 simulation. Before LNY, simulated NO$_2$ concentrations with both the baseline and adjusted emissions inventory agreed well with observations, implying that there were no significant changes in the NO$_x$ emissions level between 2016 and 2020. Near LNY, the baseline NO$_2$ simulations differ significantly from observations, while the simulation with adjusted

emissions successfully reproduced the huge reductions of the LNY and pandemic period. The difference between the baseline and adjusted simulations almost disappears at the end of March, consistent with the result of the time-series analysis (**Figure 2**). On the other hand, the baseline $SO_2$ simulations greatly overestimate observations by two or three times, implying that nominal, real-world $SO_2$ emissions in 2020 are much smaller than those reflected in the 2016 emissions inventory. By applying the top-down adjustment described here, simulations could successfully reproduce surface $SO_2$ concentrations, reducing RMSE by 93% from 9.19 to 0.62 ppb. The updated $SO_2$ and $NO_x$ emissions inventories appear to successfully reproduce variations in surface $PM_{2.5}$ concentrations, even after the start of LNY celebrations. However, in early February, as the impact of the COVID pandemic became more significant, the baseline run (with the CREATE emissions inventory) does not simulate a sudden drop in $PM_{2.5}$ observations, while the adjusted emissions run does so.

A closer look, however, reveals that the real trend in $PM_{2.5}$ emissions cannot be explained by the change of two major inorganic aerosol precursors, $SO_2$ and $NO_x$. **Figure 5** depicts the time series of normalized mean biases (NMB) of surface $PM_{2.5}$ concentrations. Before LNY, $PM_{2.5}$ NMB is mostly negative, showing the adjusted emission simulation slightly underestimates particulate matter. After LNY, $PM_{2.5}$ NMB changes prominently, showing the simulation clearly overestimates by about 20% NMB in $PM_{2.5}$ concentration. Before and after LNY, $PM_{2.5}$ NMB moves by 25.1%, from -4.1% to 21.0%, implying that the model suddenly overestimates $PM_{2.5}$ concentrations by 25% after LNY. In other words, unknown, non-modeled emissions (that is, non-$SO_2$ and non-$NO_x$ emissions) clearly reduce during the pandemic period (February and March) enough to account for 25% of total $PM_{2.5}$ concentration at baseline. This result is consistent with findings (Section 4.1) that changes in $SO_2$ and $NO_x$ emissions alone cannot explain the reduced $PM_{2.5}$ concentrations in March.

### 4.3 Sectoral contributions to emissions

One remaining question is why the recovery of $NO_x$ emissions and unchanged $SO_2$ emissions at the end of March did not lead to the recovery of $PM_{2.5}$, which might be explained by considering the time-varying emissions contribution of each economic sector. Sensitivity tests using the CMAQ model reveal that the residential and agricultural sectors are most dominant in the early months of the year (**Figure 6**), accounting for more than 60% of surface $PM_{2.5}$ concentration over China. As emissions in the residential sector are primarily from cooking and heating with anthracite coal and wood, emissions which continue even during a pandemic, one possible explanation is that emissions from the agricultural sector reduced as a result of pandemic-related delays in planting and fertilizing.

February is the start of the spring-crop planting period in southern China. The coronavirus outbreak could have impacted both field crops and livestock farms. Inputs, such as fertilizer and animal feed, have reportedly been scarce as a result of transportation disruptions, and seasonal workers have reportedly been lacking due to quarantine controls or fears (Quanying, 2020; Yu, 2020; Zhang and Xiong, 2020). Agricultural activities that generate particulate matter, such as biomass burning to clear debris and the generation of airborne dust during tilling, are reduced in intensity during the pandemic. Reduced $NH_3$ emissions as a result of diminished livestock farming activities might also be a factor leading to lower $PM_{2.5}$ concentrations.

### 4.4 Further discussions on the methods

### 4.4.1 On the data processing of time series analysis

We further discuss data-processing procedures here. **Figure 7** presents a time series of surface pollutants proceeding through data-processing steps. Even in raw format, $NO_2$ exhibits clear impacts from the pandemic. Impacts on other pollutants (CO, $PM_{10}$, and $PM_{2.5}$), however, are not easily recognizable until confounding signals are fully removed. Interpreting $SO_2$ concentration data is particularly illuminating. While 2020 $SO_2$ concentrations are substantially lower than those of previous years, the time series

obtained after the data processing described here suggests that $SO_2$ emissions are mostly consistent before and after LNY. That is, lower $SO_2$ concentrations in 2020 seem to be a continuation of year-over-year reductions and not a result of the pandemic.

Note that the various instances of linear assumptions used in this analysis should be interpreted with caution especially considering its spatiotemporal resolution and chemical characteristics. Variations in emissions and in chemical and physical processes, including chemical reactions, transport, and dispersion, can create large gradients on local scales that are likely poorly represented in the WRF and CMAQ modeling performed here, even as their importance is somewhat smoothed over regional and nationwide scales. Observed concentrations of a pollutant are generally proportional to the emissions associated with that pollutant;

conceptually, a simple linear relationship between emissions and pollutants is assumed. For the pollutants $NO_2$ and $SO_2$, these are $NO_x$ and $SO_2$ emissions, respectively. BAE2020 demonstrated that this concentration-to-emission conversion method can be used effectively at the Chinese prefecture level. Discussion of the spatial representativeness of Chinese surface-monitoring data and associated uncertainties is also presented in BAE2020. For inferring $PM_{2.5}$-related emissions, the analysis is more complicated, because $PM_{2.5}$ results from both primary and secondary (precursor) emissions. While the pollutant–emissions relation for $PM_{2.5}$ is

nonlinear, especially over relatively small spatial and temporal scales, it is still approximately valid over larger geographical regions and longer time periods.

The validity of the linear assumption was tested through a model sensitivity analysis. A CMAQ simulation with 50% reduced emissions yielded approximately 50% reduction in surface $PM_{2.5}$ concentrations over most regions in China (**Table S1**). Taken as a whole, surface $PM_{2.5}$ concentrations are roughly proportional to overall emissions. Thus, the simplifying assumption of linearity

appears reasonable for the more complex $PM_{2.5}$ case, generating a time series of estimated pollutant emissions without meteorological variations. Nevertheless, $PM_{2.5}$ emissions estimated with this analysis are necessarily more uncertain than are $NO_x$ emissions. Notably, **Table S1** also shows that CMAQ simulations with adjustments in $SO_2$, $NO_x$, and $NH_3$ individually showed disproportionately lower responses, suggesting that surface $PM_{2.5}$ concentrations are influenced by other emissions (e.g., elemental carbon and organic carbon emissions) and/or nonlinear processes that likely vary with atmospheric chemistry regime.

**4.4.2 On the emissions adjustment experiment**

As stated in the methodology section, we further discuss here the emissions-to-concentration sensitivities (i.e. $\beta$). The $\beta$ values can be calculated using any two model simulations based on different emissions inputs, by comparing the change in emissions with the change in simulated concentrations. Furthermore, if we specifically change the emissions according to the ratio of observations and the base model simulation, we further simplify the emissions scaling factor as follows.

For this simulation, *adj1*, if we apply the adjusted emissions using the ratio of the observed and modeled concentrations, the adjusted emissions for the *adj1* run, $E_{adj1}$, are

$$E_{adj1} = \frac{C_{obs}}{C_{base}} \cdot E_{base} \qquad (6)$$

If we apply this to Eq. (4), we can obtain

$$\beta = \frac{E_{adj1}/E_{base}}{C_{adj1}/C_{base}} = \frac{C_{obs}/C_{base}}{C_{adj1}/C_{base}} = \frac{C_{obs}}{C_{adj1}} \qquad (7)$$

Therefore, the emission adjustment factors in the next simulation (*adj2*) can be found using Eq. (5):

$$E_{adj2} = \beta \cdot \frac{C_{obs}}{C_{base}} \cdot E_{base} = \left[\frac{C_{obs}}{C_{adj1}} \cdot \frac{C_{obs}}{C_{base}}\right] \cdot E_{base} \qquad (8)$$


where *adj2* indicates the second and final simulation for the top-down emissions adjustment method.

From here, the $\left[\frac{C_{obs}}{C_{adj1}}\right]$ term, or $\beta$, can be interpreted as an additional adjustment factor to the original adjustment factor in *adj1*, $\left[\frac{C_{obs}}{C_{base}}\right]$. If the emissions modification in *adj1* results in the same percentage change in concentrations, $C_{obs} / C_{adj1} = 1$, we do not need the secondary adjustment. If the simulated concentration from *adj1* is smaller (larger) than the observations, we need to

increase (reduce) the amounts of emissions. This procedure was applied to create new 2020 emissions of both $SO_2$ and $NO_x$.

In most cases, the calculated $\beta$ values are close to one (**Figure S5**), implying that the simple assumption $\beta = 1$ in BAE2020 remains effective. The $\beta$ values for $NO_x$ emissions are slightly higher than those for $SO_2$ emissions over polluted areas (**Figure S6**), which implies that more secondary reactions are involved in tropospheric $NO_x$ chemistry.

Both enhancements to the top-down simulations—$\beta$ values and the daily application of emission adjustment factors—clearly

improved the model's performance, especially in the pre-LNY periods. While the monthly emissions adjustments failed to represent the rapid changes in $NO_2$ concentrations after January 25, 2020 (**Figure S7**), the daily adjustment method successfully modeled these changes (**Figure 4**). The general underestimation of $NO_2$ concentrations was corrected using the $\beta$ values (**Figure 4**). The improved model performance was confirmed by comparing the spatial distributions and scatterplots before and after these adjustments (**Figures S8–S10**). Spatial distributions of RMSEs of model performances in $SO_2$, $NO_2$ and $PM_{2.5}$ are also summarized

in **Figure S12**.

Understanding the characteristics of the $\beta$ values in terms of their spatial distribution, temporal variation, and chemical difference is important for several reasons. In the emission update procedure in practice, we can apply the pre-calculated $\beta$ values from the look-up table if the $\beta$ values show general consistency according to their location, time, and chemical component. For the emission control policy, the $\beta$ values provide valuable information on the efficiency of emissions control because they suggest how

effectively pollutant concentrations can be removed given the amount of emissions control by the government.

**Figure 8** summarizes the characteristics of the $\beta$ values. As they are defined as the ratio of the emissions change (i.e. $E_{adj1} / E_{base}$) to the change in concentrations (i.e. $C_{adj1} / C_{base}$), the slopes of the fitted lines in the scatterplots describe the emissions-to-concentration sensitivities for $SO_2$ and $NO_2$ (**Figure 8a & b**). The histogram of the occurrence of the $\beta$ values also confirms that for both $SO_2$ and $NO_2$, the calculated $\beta$ values are centered slightly over one (mean=1.42 and median=1.27 for $SO_2$ and mean=1.40

and median=1.26 for $NO_2$) (**Figure S13**). **Figure 8c & d** demonstrate the spatial distributions of the $\beta$ values over Chinese territories. Except a few outside locations, the $\beta$ values are mostly consistent, around one. We further investigated the temporal variations of the $\beta$ values by showing the daily variations of the estimated $\beta$ values for selected Chinese provinces (**Figure 8e & f**). It is evident that the $\beta$ values differ by location, implying that the emissions-to-concentration sensitivities vary for different regions likely due to their unique chemical and emission environment. However, for each location, the $\beta$ values are mostly

consistent over time. For the practical use of the $\beta$ values in the emission update procedure, we may use region-specific sensitivity parameterization since their temporal variations over a specific region are not significant.

To evaluate the emissions update approach, the key feature in this study is the validation of $PM_{2.5}$ concentration. We used observation-based $SO_2$ and $NO_2$ emissions adjustments and there was no adjustment in the primary $PM_{2.5}$ emissions, meaning that the improvement of $PM_{2.5}$ is achieved through chemical reactions and their balances. The surface concentrations of surface $PM_{2.5}$

concentrations, especially inorganic aerosols, are formed by secondary reactions, which are determined by the balance of chemical

reactions for nitrate, sulfate, and ammonium. The performance of the $PM_{2.5}$ simulations provides strong evidence that the top-down emissions adjustment method used in this study is valid and successfully reproduces a realistic chemical environment.

Formation efficiency of sulfate aerosols by updating $SO_2$ and $NO_x$ emission is also very interesting. From Figure 4, one may notice that the change of total $PM_{2.5}$ concentration is not prominent in the pre-pandemic period, even with strong reduction in $SO_2$

emissions. Modelled PM speciation components show that the reduced sulfate concentrations were cancelled out by the increased nitrate concentrations, due to the balance of non-linear nitrate-sulfate-ammonium chemistry. Nitrate is the most dominant component of $PM_{2.5}$ during the wintertime (contributing ~50% while sulfate contributes 14%), and the sudden drop of $PM_{2.5}$ concentrations during the pandemic is mostly driven by the change of nitrate concentrations. This result implies an important message to emissions control policy, suggesting that both $SO_2$ and $NO_x$ emissions reductions will be required to achieve better

emission reduction efficiency.

## 5. Summary

We investigated changes in observed surface-pollutant concentrations and precursor emissions over China and inferred changes in human activity as a result of the coronavirus pandemic. Three analyses were conducted: (1) a time series analysis, (2) emissions adjustment experiment, and (3) sectoral emission contribution estimations. First, we removed four types of variation

(meteorological, weekly, yearly, and the LNY) to isolate impacts of coronavirus pandemic from observed surface pollutant concentrations. A chemistry model simulation with fixed emission inventory was used to remove meteorological variations. The analysis has shown that $NO_x$ emissions across China recovered to almost normal levels two months after LNY. However, considering the estimated changes in emissions associated with $PM_{2.5}$, some emissions remain missing, as of the end of March 2020, compared with normal years. Second, an alternative modeling approach using updated real-time $SO_2$ and $NO_x$ emissions

also suggested that about 25% of $PM_{2.5}$ emissions are likely missing from the period. Third, impacts of sectoral emissions were presented to infer the role potential missing emissions or activities.

The surface observations of pollutants and inferred precursor emissions across China suggest that the country is greatly recovering, as evidenced by the apparent resumption of near-normal transportation-related emissions. The pandemic appears not to have strongly affected the industrial sector; continued depression in estimated $PM_{2.5}$-associated emissions may be due to effects on the

agricultural sector. If the sustained reduction in $PM_{2.5}$ is due to reduced activity in the agricultural sector, agricultural production could be affected, at least in the short term. This could hold important implications for China's path to recovery and, potentially, for broader parts of the world, if similar types of agricultural impacts occur elsewhere.

The data analysis approach used here has attempted to isolate the ambient data signal due to the coronavirus from other sources of variation. The apparent difference between the recovery timelines for $NO_2$ and $PM_{2.5}$ suggests that estimating $NO_x$ emissions alone

is insufficient to draw conclusions about the overall recovery of the Chinese economy. Overall, changes in concentrations of atmospheric pollutants can provide useful information about the spatial and temporal economic impacts of the coronavirus pandemic, a serious global issue.

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

**Acknowledgements**

This research is supported by the National Strategic Project-Fine Particle of the National Research Foundation of Korea (NRF) funded by the Ministry of Science and ICT (MSIT), the Ministry of Environment (ME), and the Ministry of Health and Welfare (MOHW) (2017M3D8A1092015).

**Disclaimer**

The scientific results and conclusions, as well as any views or opinions expressed herein, are those of the author(s) and do not necessarily reflect the views of NOAA or the Department of Commerce.

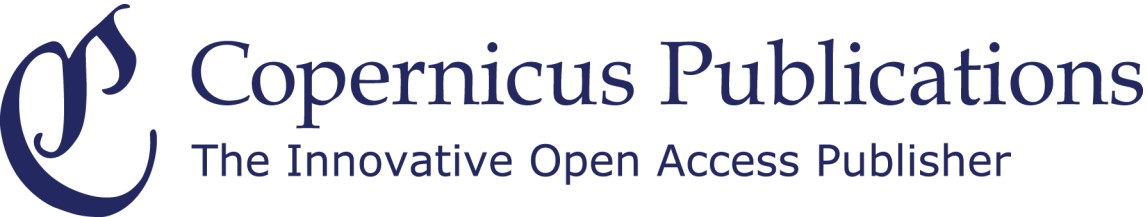


**Table 1. Physical options for meteorological and chemical simulations**

| Model | Physical options | Descriptions |
|---|---|---|
| WRF v3.4.1 | Initial field | FNL (NCEP, 2000) |
| | Microphysics | WSM6 (Hong et al., 2004) |
| | Cumulus scheme | Kain-Fritsch (Kain, 2004) |
| | Land surface model scheme | NOAH (Chen and Dudhia, 2001) |
| | Planetary boundary layer scheme | YSU (Hong et al., 2006) |
| CMAQ v4.7.1 | Chemical mechanism | SAPRC99 (Carter, 2003) |
| | Chemical solver | EBI (Hertel et al., 1993) |
| | Aerosol module | AERO5 (Binkowski and Roselle, 2003) |
| | Advection scheme | YAMO (Yamartino, 1993) |
| | Horizontal diffusion | Multiscale (Louis, 1979) |
| | Vertical diffusion | Eddy (Louis, 1979) |
| | Cloud scheme | RADM (Chang et al., 1987) |


**Table 2. Comparison of data-processing steps in the emissions-adjustment methods used in BAE2020 and this study**

| Data-processing steps | BAE2020 | This study |
|---|---|---|
| Spatial processing | Prefecture-level | Prefecture-level |
| Temporal processing | Monthly | Daily (14-day moving average) |
| Emission-to-concentration conversion factor ($\beta$) | $\beta = 1$ | Varying (Daily & prefecture-level) |
| CMAQ simulations | 1 (adj1) | 2 (adj1 & adj2) |
| Emissions adjusted | $SO_2$ | $SO_2$, $NO_x$ |
| Note | | Results of 'adj1' simulations were used to calculate beta values for 'adj2' simulation |

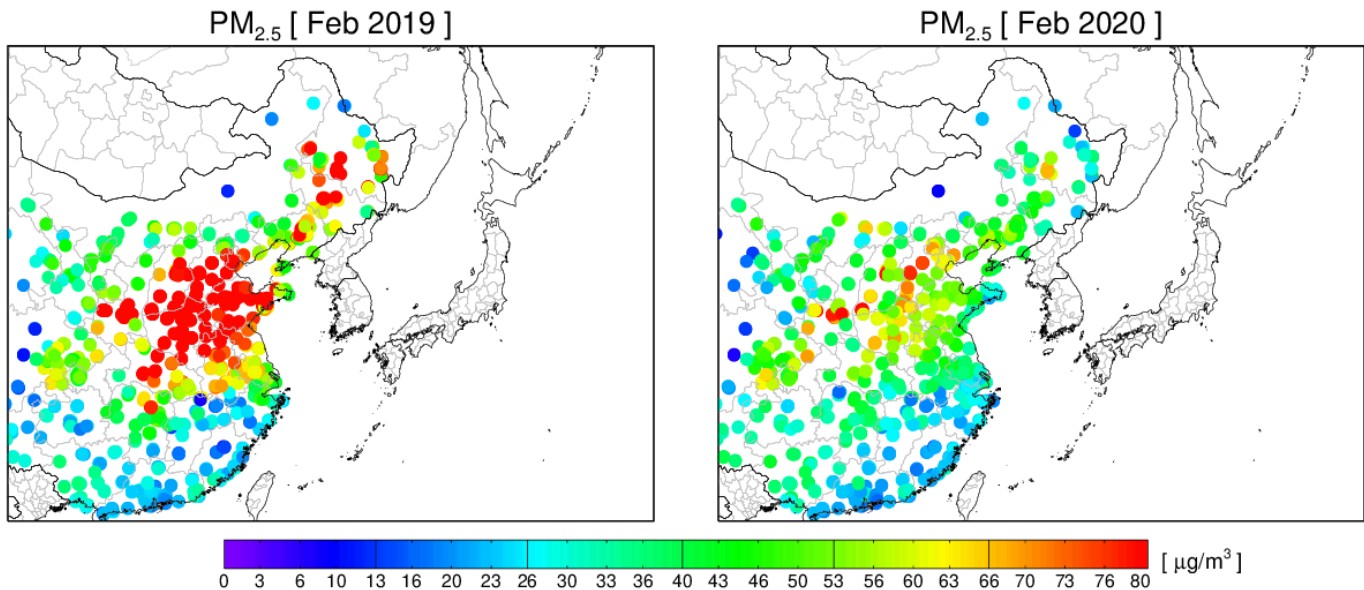

**Figure 1. Geographical coverage of modeling domain and surface-monitoring sites. Monthly mean surface PM$_{2.5}$ concentrations in February 2019 and February 2020 are shown.**

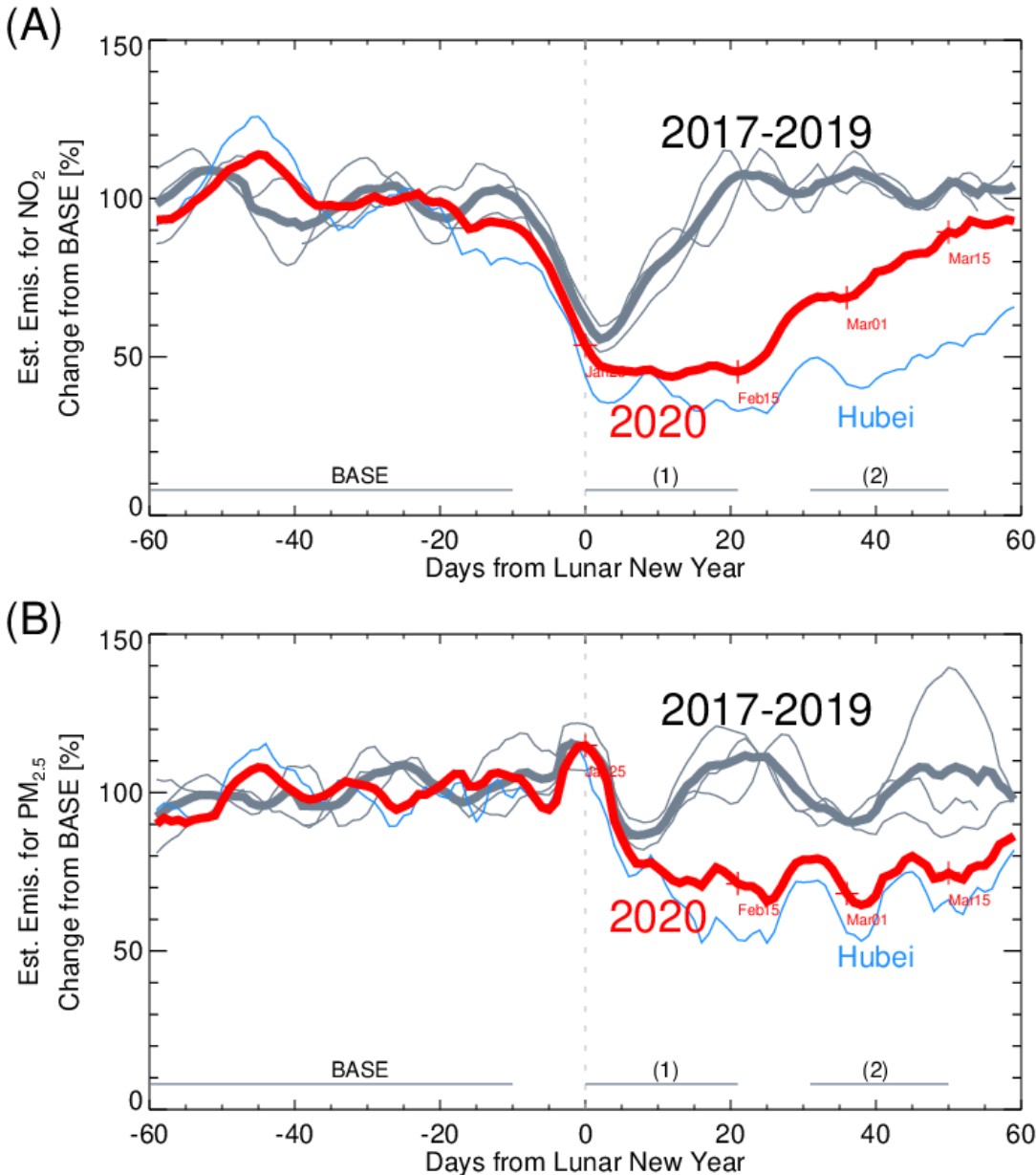

**Figure 2. Time series of estimated emissions for (a) NO₂ and (b) PM₂.₅ using 1,332 surface monitoring sites across China. The gray lines indicate 2017–2019 variations, with their average in the thick gray line, whereas the red line indicates the 2020 variation. The blue line indicates the 2020 variations in Hubei province (46 sites). BASE is used as the pre-LNY period, and (1) and (2) denote the period of maximum impact and the recovery period, respectively.**


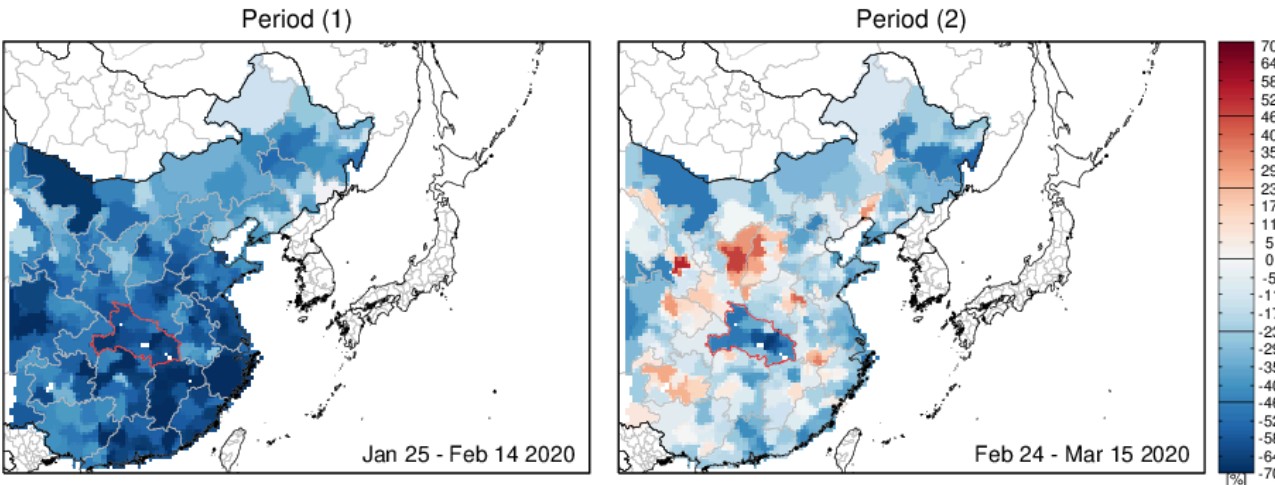

**Figure 3. Spatial distribution of the change in estimated NOx emissions from the baseline period (Figure 2) during the period of maximum impact (January 25 – February 14, 2020) and the recovery period (February 24 – March 15, 2020). Hubei province is marked in red.**


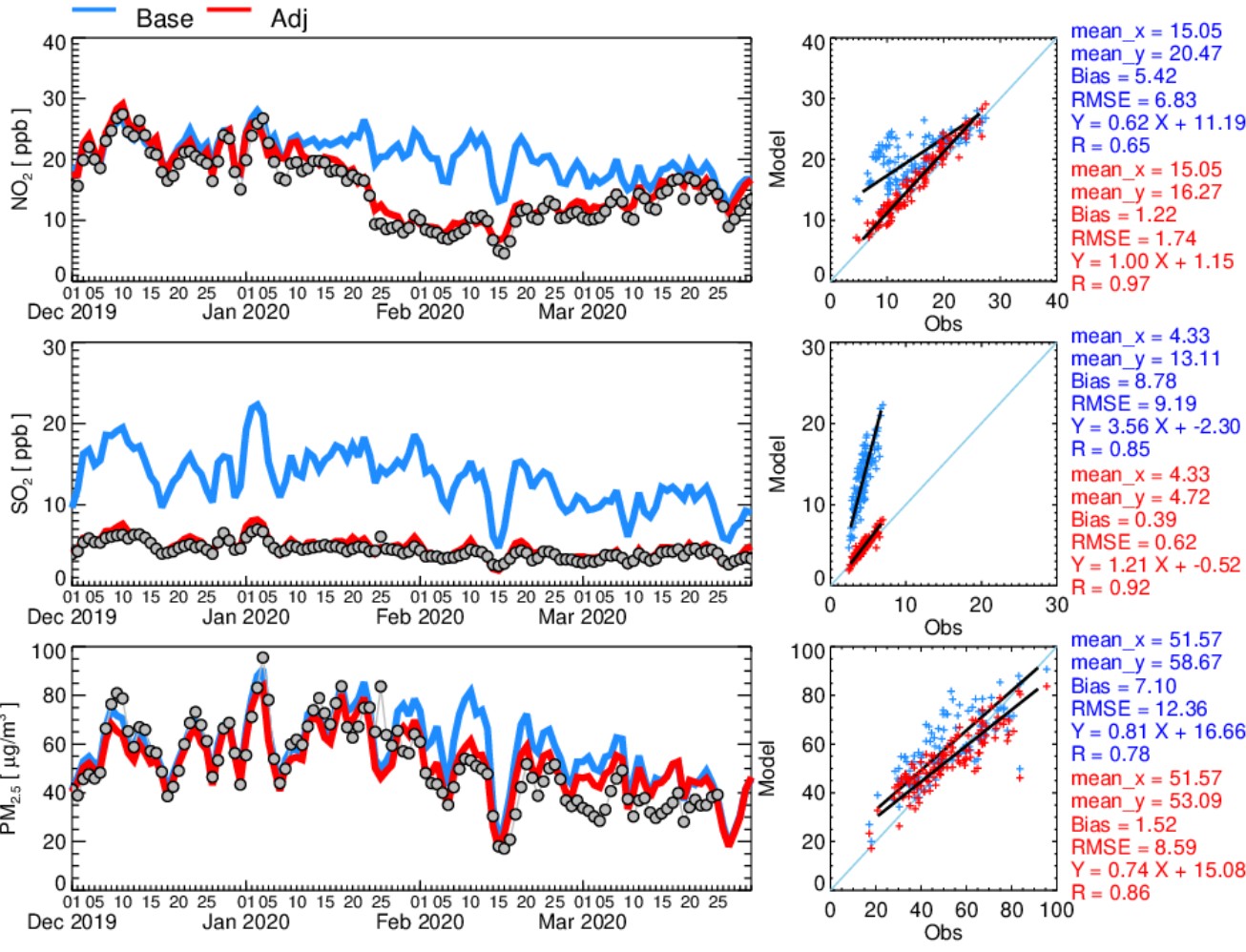

**Figure 4. Time series and scatter plots of observed and modeled surface concentrations of SO₂, NO₂, and PM₂.₅ from 1,332 Chinese surface-monitoring sites during the pandemic period. Model simulations using the baseline emissions inventory (CREATE) and top-down adjusted emissions are shown in blue and red, respectively. Observations are represented by gray circles.**


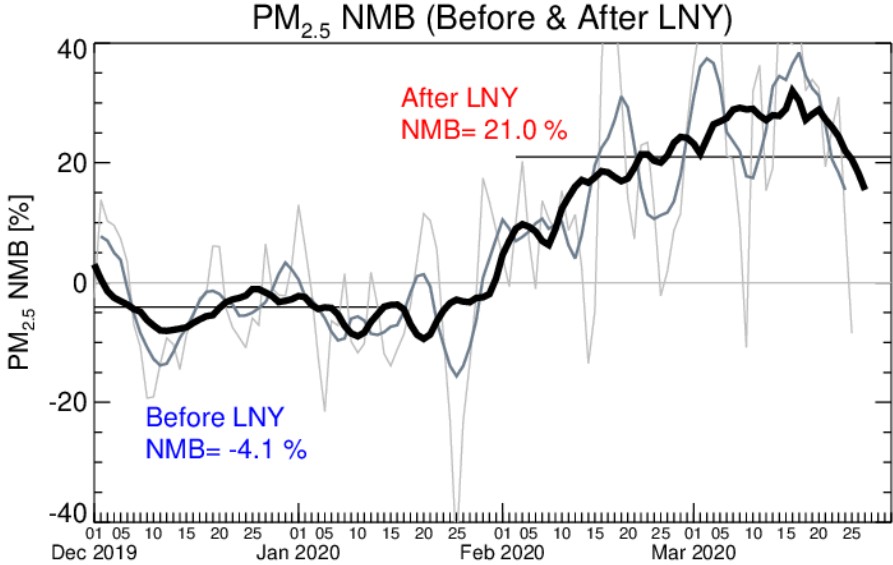


**Figure 5. Time series of surface PM₂.₅ normalized mean bias during the pandemic period between observed and modeled data with adjusted emissions (i.e., SO₂ and NOₓ emissions adjusted). Mean NMB before and after LNY are also marked. Raw, 7-day, and 14-day moving average NMBs are shown in thin, medium-thin and thick lines, respectively.**

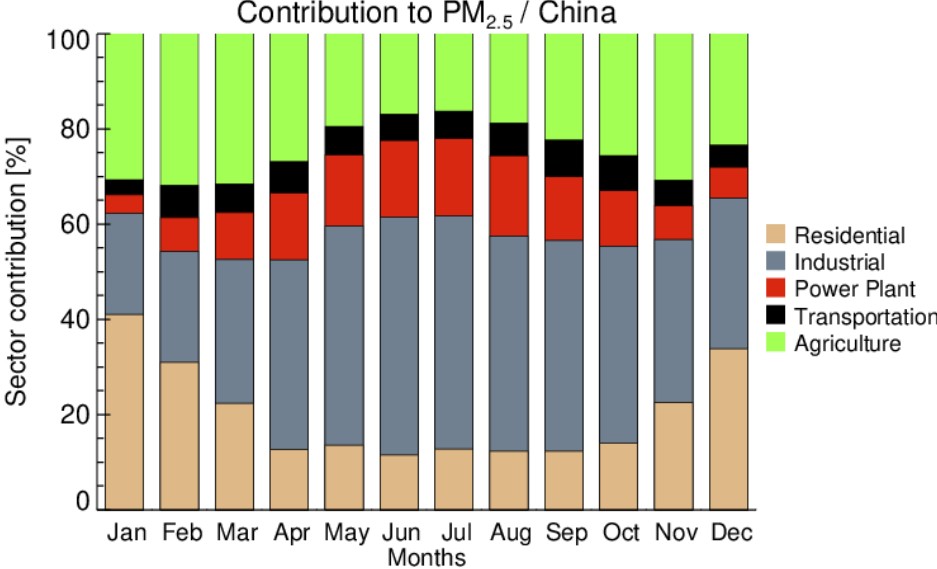


**Figure 6. Monthly variations in emission contributions to surface PM₂.₅ concentrations over China by sector. The contributions from the five sectors (residential, industry, power generation, transportation, and agriculture) were estimated using a brute force perturbation method.**

DATA PROCESSING PROCEDURES

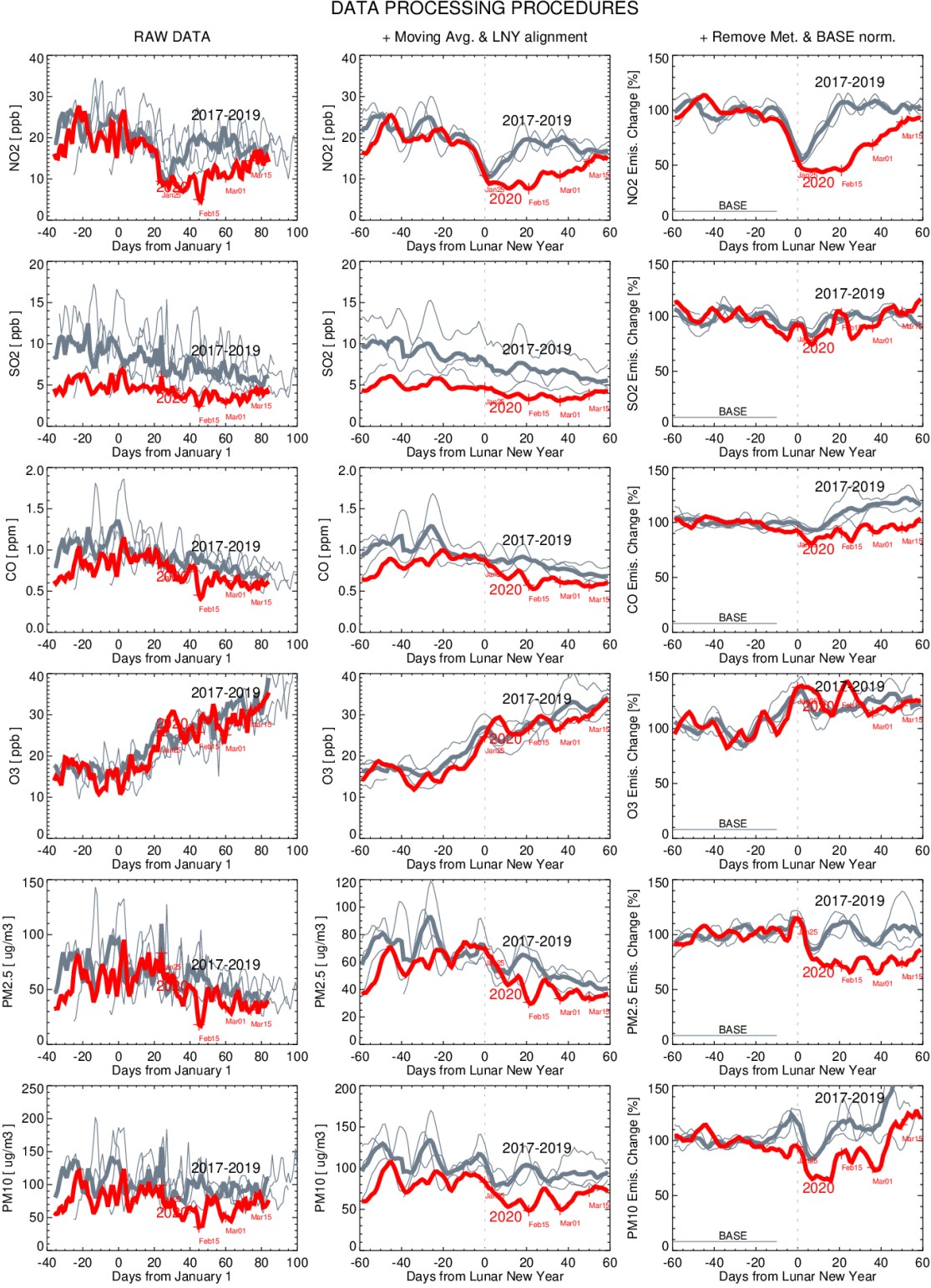

Figure 7. Time series of surface NO₂, SO₂, CO, O₃, PM₂.₅, and PM₁₀ concentrations over China following the data-processing procedures step by step. Raw data (left column), data after applying a seven-day moving average and an LNY alignment (middle column), and data after removing meteorological variations and calculating variations from the baseline periods (right column) are all shown.


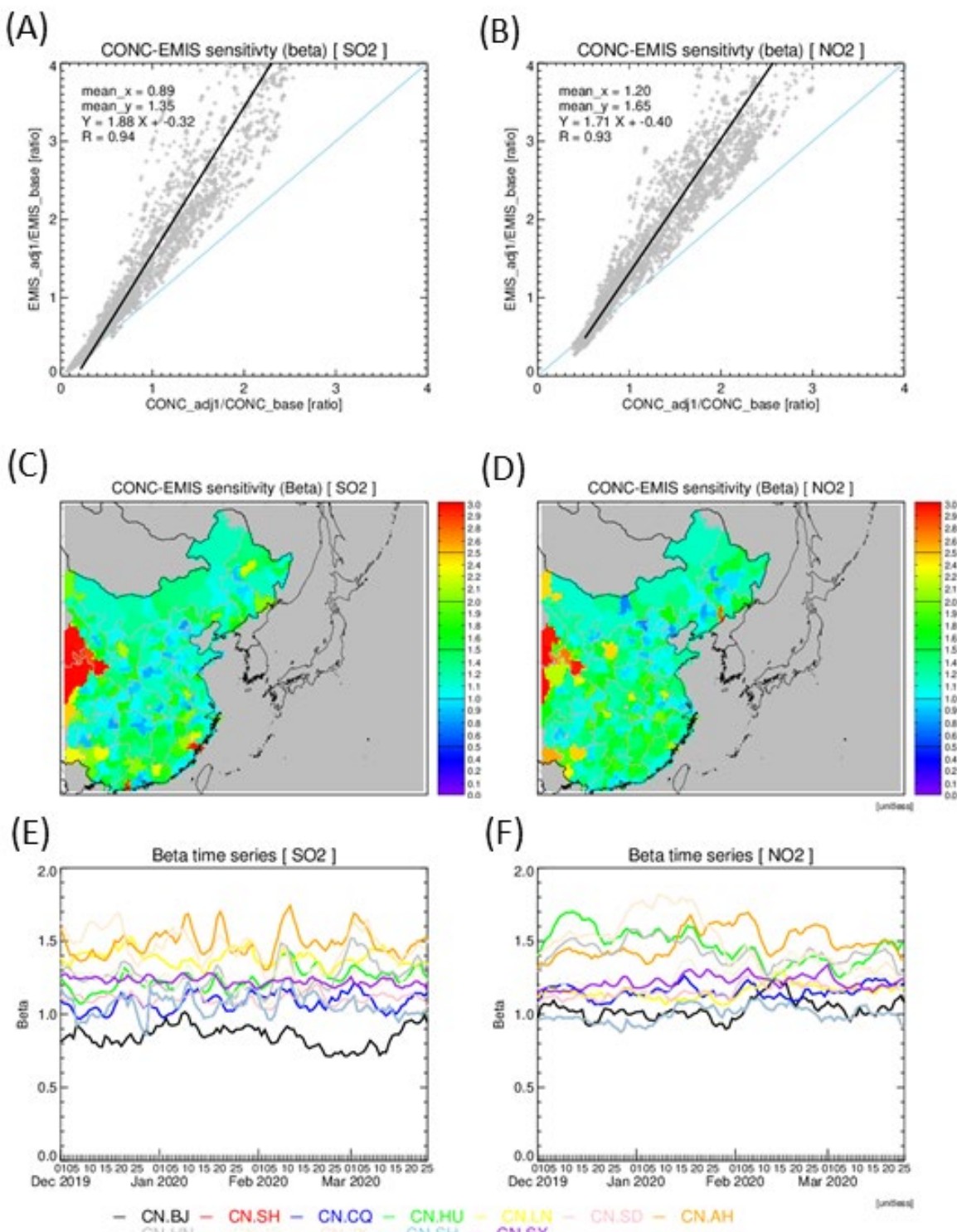


**Figure 8 Calculation of the concentration-to-emissions sensitivities (β) for the emissions adjustment experiment of SO₂ (left column) and NO₂ (right column). The β values are obtained as the ratio of the emissions change (i.e. Emis_adj/Emis_base) to the change in concentrations (i.e. Conc_adj1/Conc_base), which is also consistent with the slope in the scatterplot (A & B). Spatial variations of the average concentration-to-emissions sensitivities (β) during January to March 2020 over China (C & D). The temporal variations of the**

**β values for selected Chinese provinces are shown in the lower panel (E & F). (BJ=Beijing, SH=Shanghai, CQ=Chongqing, HU=Hubei, SD=Shandong, AH=Anhui, HN=Hunan, JS=Jiangsu, SX=Shanxi).**