# Peer review of "Quantitative assessment of changes in surface particulate matter concentrations and precursor emissions over China during the COVID-19 pandemic and their implications for Chinese economic activity"

_Atmospheric Chemistry and Physics, 2020_

## Referee Comment (RC1) · Anonymous Referee #1 · 4 Sep 2020

This paper applies interesting methods that combine atmospheric chemistry transport modelling with surface observations to quantify changes in concentrations and emissions of SO2, NOx and PM2.5 across (most of) China during the COVID-19 lockdown and the recovery period immediately following coming out of the full lockdown. Simulations were conducted based on both a bottom-up inventory and top-down assimilated emissions. The methods applied need to disentangle the coincidence of the lockdown with the Chinese Lunar New Year holiday (which itself impacts on air pollution levels and has different dates in different years) and the impacts of meteorology, day-of-week,

and year-on-year trend.

The authors show that SO2 emissions were little effected during the lockdown, that NO2 emissions started to recover fairly quickly, but that PM2.5 still remained about 30% lower than expected at the end of March. The authors suggest this is due to substantial depression and delays in agricultural activities during and immediately post the lockdown, including reduced dust emissions from tilling, reduced biomass burning particles and some reduced NH3 emissions. The point is well made that changes in air quality and in emissions from COVID-19 lockdown cannot be generalised from examination of just one pollutant.

The paper is succinctly and clearly presented. The figures are similarly well presented and captioned and plenty of supporting analyses are presented in a comprehensive supplementary information. It is welcome to read a manuscript on the impacts of the pandemic on air quality that applies proper quantitative methods to try and overcome confounding factors on air quality.

A few minor comments:

L43: Surely residential, power generation and industry are all also major sources of NOx in China, such that transportation is not the majority source in many areas?

L48: Surely for the time duration being investigated here (weeks), meteorologically-driven variation in pollutant concentration is very important, and more important than natural inter-annual variations?

L50: Yes, here the text does refer to meteorological variations so amend the phrasing of the equivalent point a couple of sentences earlier.

L67: State the time period or periods over which the 80% data availability criterion was applied.

Figure 1: Please increase the font size on the figure legend.

[Figure]

---

## Referee Comment (RC2) · Anonymous Referee #2 · 28 Sep 2020

In the manuscript, the authors used 3-D chemical transport model along with other surface and remote measurements to study the change of surface concentrations in China due to the COVID-19 pandemic. One of their important strategies is to adjusting emission inventory which they claimed to be reasonable. I think their data analyses are reasonable. Still, I am not comfortable with their approach of adjusting emission inventory. I suggest the authors should provide more detailed explanation on it and how they could validate it. One more thing that concerns me is that it seems like that the manuscript does not contain significant scientific finding(s). It shows just the trend

and some implications. Please provide why the results shown in the manuscript are important. Still, the method used in the manuscript looks interesting and I recommend the authors have more explanation and provide what are the important findings in this study. To sum it up, I recommend the manuscript should revised significantly and submitted again.

---

## Referee Comment (RC3) · Anonymous Referee #3 · 28 Sep 2020

This study intends to investigate the timeline of economy recovery in China using the changes in emissions of aerosol precursors during the COVID-19 pandemic, 2020. Since the emissions highly depend on the human and economic activities, an accurate assessment on emissions in China may provide some proves for China economy in that period. Without the bottom-up emission inventory in 2020, this study estimates the emissions of NOx, SO2 and other 4 kinds of pollutants (Fig. S3) by combining the WRF/CMAQ simulations and observed concentrations of NOx, SO2 and aerosols. Inventory in 2016 is used and adjusted to 2020 after comparing the simulated and

observed pollutant concentrations. Then plenty of analyses are performed according to the estimated emissions. Overall, the paper is well structured and provides abundant results on the topic. However, there are several noticeable flaws in the manuscript. Hence, these analytic results would be questioned.

1. I do know why the authors named the paper as "Quantitative assessment of changes in surface particulate matter concentrations...", since the study mainly talks about the variations in emissions using model results and an emission-adjustment method. The comparison on surface particulate matter concentrations between the period and previous years has been showed by other studies or reports as the study mentioned. And we scarcely need a model assessment when we have the pollutant observations, considering the model results were adjusted by observations in the manuscript.

2. The study claimed that "Meteorological influences were reduced by combining surface data with output from a three-dimensional chemistry model to calculate estimated emissions" (Section 3.1). I do not understand why combining the pollutant observations with model simulation can reduce the meteorological effects. The concentration time series, no matter from observation or from model, would be varied with the simulated/realistic meteorology. And, the adjusted emissions computed using eq. (1) or (6) should change with varied pollutant observations following meteorology.

3. Again, I am afraid that the "less sensitive of emission to meteorological variations" (line 119 and Fig. 2) would be not related to the combination process but due to the smoothing process with 7-day running average. If there is no any smoothing process, I believe the estimated emission time series would variation more sharply than Figure 2 shown. Hence, the combination based on linear ratio of concentrations could not remove the meteorological influences on emission estimation.

4. Top-down emission adjustments is the kernel of this study (Section 3.2). Here ratio between observed and simulated pollutant concentrations in every grid cell and day are used as a base scaling coefficient (eq. (3)). Accurate emission estimation using the eq.

(3) strongly depends on the simulation quality in COVID-19 pandemic. However, the study only shows a daily time series validation during whole years (I guess the domain-averaged concentrations. The paper did not mention). It is not enough for this study. The authors should provide an estimation in every grid-cell (or sites) near LNY-period.

5. The study also introduces another coefficient $\beta$ to furtherly adjust the ratio between observed and simulated pollutant concentrations. However, the major flow is that the study determines the $\beta$ using two simulated concentration (adj1 and base in eq. (5)). Here the $\beta$ in eq. (5) reflects the scaling relationship between simulation "adj1" and "base" but not the relationship between a simulation and an observation. In another word, the $\beta$ in eq (5) should not be the one in eq. (3). Therefore, the introducing process of $\beta$ in eq. (5) "as Equation (3) can be written (line 152)" are not reasonable.

6. The study used the pollutant observation in the COVID-19 pandemic to validate the emission adjustment method (line 169). But considering the emission adjustment process is determined by the pollutant observations in the same period and grid-cells and $\beta$ almost equal to 1, it is not a meaningful validation for the method.

---

## Author Comment (AC1) · 24 Nov 2020

Authors' responses to reviewer comments

**"Quantitative assessment of changes in surface particulate matter concentrations over China during the COVID-19 pandemic and their implications for Chinese economic activity" by Kim et al.**

**General response**

Authors express their appreciation to the three reviewers and the editor. Thanks to their productive comments, we were able to improve our manuscript. We provide below the general responses and the point-by-point responses to the reviewer's comments. Reviewers' comments are shown in italics.

Here are three major points in the responses to the reviewers' comments.

1.  Two independent analyses

In this study, we provide two analyses. They are independent to each other, but suggest a consistent conclusion at the end. The time series analysis (Section 4.1) uses a bottom-up emissions inventory, and the top-down emissions adjustment experiment (Section 4.2) uses top-down emissions inventory. While both analyses reach to the same conclusion (i.e. ~30% missing of PM2.5 emissions, potentially agricultural emissions), we never mixed up their results. Two analyses were conducted independently.

2.  Significance of analysis

In terms of regional air quality, COVID-19 pandemic is a rare opportunity of large-scale natural experiment for emissions control. This study suggests a novel method to update near real-time NOx and SO2 emissions, and it demonstrates that this method works efficiently by comparing with actual observations.

In terms of COVID-19 pandemic, this study demonstrated that the change of economic sectors can be estimated based on the pollutant concentrations. Several important conclusions were suggested, including –

(1) To assess quantitatively the changes in emissions, we need to isolate confounding factors from meteorological, emissions control, and socioeconomic (e.g. LNY) factors.

(2) Different recovering speeds for different economic sectors, so changes in air quality and in emissions from COVID-19 lockdown cannot be generalized from examination of just one pollutant.

(3) There are potential missing emissions precursors for PM2.5. This study suggests potential lack of agricultural activity. If true, it has crucial implications in agricultural production.

3.  Validation of top-down emission adjustment

We provide improved statistics by using updated NOx and SO2 emissions. While we do understand reviewer's concern about potential self-validation, especially for SO2 and NO2 concentrations. However, the key feature in this study is the validation of $PM_{2.5}$ concentration. We used observation-based SO2 and NO2 emissions adjustment. PM2.5 is totally independent. By updating SO2 and NOx emissions, the model was able to reproduce the changes of PM2.5 successfully. This result confirms the importance of inorganic aerosols in current PM2.5 concentrations. This is a strong evidence showing that the top-down emissions adjustment method used this study is valid.

We also provide point-by-point responses to reviewers' comments.

Reviewer #1

*L43: Surely residential, power generation and industry are all also major sources of NOx in China, such that transportation is not the majority source in many areas?*

Thanks for the comment. Transportation, power generation and industry are three major NOx emissions sources in Asia (Li et al., 2017). In China, the industrial sector is the most dominant sector, followed by power generation and transportation in the year of 2010 (Li et al., 2017). In the manuscript, we stated the role of transportation sector since it represents characteristics of urban anthropogenic emissions well. We also believe that the role of transportation sector in NOx emissions will be prominent after long-term efforts of emissions control in industry and power generation, and with rapid growth of mega cities.

[Figure]

**Figure 4.** Emission distributions among sectors in Asia in 2010.

| Regions | SO$_2$ | NO$_x$ | CO | NMVOC | NH$_3$ | PM$_{10}$ | PM$_{2.5}$ | BC | OC | CO$_2$ |
|---|---|---|---|---|---|---|---|---|---|---|
| **China** | 28 663 | 29 071 | 170 874 | 23 619 | 9804 | 16 615 | 12 200 | 1765 | 3386 | 10 124 |
| Power | 8166 | 9455 | 2077 | 255 | 0 | 1389 | 893 | 2 | 0 | 3245 |
| Industry | 16 775 | 11 218 | 71 276 | 14 461 | 239 | 9451 | 6061 | 575 | 530 | 4928 |
| Residential | 3489 | 1140 | 76 579 | 6349 | 450 | 5246 | 4737 | 908 | 2752 | 1266 |
| Transportation | 234 | 7257 | 20 942 | 2553 | 76 | 529 | 509 | 281 | 104 | 684 |
| Agriculture | | | | | 9040 | | | | | |

(Li et al. (2017) Table 4)

*L48: Surely for the time duration being investigated here (weeks), meteorologically-driven variation in pollutant concentration is very important, and more important than natural inter-annual variations?*

Thanks for the comment. We modified the manuscript to clarify three major components: (1) natural variations, (2) emission control, and (3) sporadic socioeconomic events. Natural variations include impacts from short term synoptic weather, interannual meteorological variations, and long-term climate change.

*L50: Yes, here the text does refer to meteorological variations so amend the phrasing of the equivalent point a couple of sentences earlier.*

Thanks for the comment. We agree and have revised the manuscript. Please, see the response above.

*L67: State the time period or periods over which the 80% data availability criterion was applied.*

Thanks for the comment. For each year we used observational data during LNY-60 days to LNY+60 days. We selected sites with more than 80% data availability for each year (2017-2020).

*Figure 1: Please increase the font size on the figure legend.*

Thanks for the comment. We have updated Figure 1.

Reviewer #2

*In the manuscript, the authors used 3-D chemical transport model along with other surface and remote measurements to study the change of surface concentrations in China due to the COVID-19 pandemic. One of their important strategies is to adjusting emission inventory which they claimed to be reasonable.*

Thanks for the comment. This study offers two independent analyses that suggests the same conclusion. Please, refer the general response 1.

*Still, I am not comfortable with their approach of adjusting emission inventory. I suggest the authors should provide more detailed explanation on it and how they could validate it.*

Thanks for the comment. We provide a strong validation for the top-down emissions adjustment method in the comparison of PM2.5 concentrations. Please, refer the general response 3.

*One more thing that concerns me is that it seems like that the manuscript does not contain significant scientific finding(s). It shows just the trend and some implications. Please provide why the results shown in the manuscript are important.*

Thanks for the comment. Please, refer the general response 2 for the significance of the study.

Reviewer #3

*1. I do know why the authors named the paper as "Quantitative assessment of changes in surface particulate matter concentrations...", since the study mainly talks about the variations in emissions using model results and an emission-adjustment method. The comparison on surface particulate matter concentrations between the period and previous years has been showed by other studies or reports as the study mentioned. And we scarcely need a model assessment when we have the pollutant observations, considering the model results were adjusted by observations in the manuscript.*

Thanks for the comment. I guess "I do know" in the reviewer's comment is a typo of "I don't know".

In a regional air quality problem, especially in the effect of anthropogenic emissions, the main cause and effect are the relationship between 'human activity', 'anthropogenic emissions' and 'pollutant concentrations'. The bottom line is that we like to infer the amount of emissions change and related economic activities, based on the actual measurements of pollutant concentrations.

Since the 'concentration-emissions relationship' depends strongly to the meteorological condition, one should consider the variations of meteorology and chemical reactions within to claim the 'quantitative assessment' of emissions change or human activity. Simple comparison of concentration time series (as demonstrated in many previous studies) cannot do that.

We claim that this study is a quantitative assessment because we tried to remove confounding factors by combining the information available from the chemistry transport model and also by applying data processing skills to remove variations from weekly, yearly and LNY effects.

Please, note that this study offers two independent analyses. Both analyses suggest a consistent result. We did not mix the analyses as the reviewer may think. They are independent. Please, also refer the general response 1.

*2. The study claimed that "Meteorological influences were reduced by combining surface data with output from a three-dimensional chemistry model to calculate estimated emissions" (Section 3.1). I do not understand why combining the pollutant observations with model simulation can reduce the meteorological effects. The concentration time series, no matter from observation or from model, would be varied with the simulated/realistic meteorology. And, the adjusted emissions computed using eq. (1) or (6) should change with varied pollutant observations following meteorology.*

Thanks for the comment. The basic concept of the time series analysis in this study is that we need to retrieve "emissions" information out of measured "concentrations". Given the same amount of emissions, the actual measured pollutant concentration would be significant different based on the meteorological condition, especially in the formation of secondary pollutant. We combine 'observed concentrations' and 'the concentration-emission relation' from model to estimate the emissions variation in the real world.

Please, also be noted that we did not use the adjusted emission simulation for time series analysis. They are separate analyses. Please, see the general response 1.

*3. Again, I am afraid that the "less sensitive of emission to meteorological variations" (line 119 and Fig. 2) would be not related to the combination process but due to the smoothing process with 7-day running average. If there is no any smoothing process, I believe the estimated emission time series would variation more sharply than Figure2 shown. Hence, the combination based on linear ratio of concentrations could not remove the meteorological influences on emission estimation.*

Thanks for the comment. We do not agree that 7-day running averaging process is the only dominant data processing step. Changes of time series analyses after the 7-day moving average and after excluding meteorological impact are already demonstrated in Figure 7. Clearly, both data processing procedures are important to capitalize the impact of the pandemic.

The 7-day moving average process was applied to remove unfair comparison by comparing different day of week. Since we applied an alignment to center the LNY, daily time series comparisons are done for different weekdays in different years because LNY days assigned to different weekday as

shown below (See table below). As the weekly variation of anthropogenic emissions is a dominant feature in anthropogenic emissions, we removed its noise by applying 7-day moving average. If we simply compare time series without removing weekly variation, we compare signals from different weekdays. Since anthropogenic emissions have prominent weekly variations, this comparison can cause unfair comparison. We do not conclude emissions changes by comparing emissions on Wednesday to emissions of Sunday. It is not a fair comparison.

| Year | Lunar New Year | Weekday |
|------|----------------|---------|
| 2017 | January 28 | Saturday |
| 2018 | February 16 | Friday |
| 2019 | February 5 | Tuesday |
| 2020 | January 25 | Saturday |

*4. Top-down emission adjustments is the kernel of this study (Section 3.2). Here ratio between observed and simulated pollutant concentrations in every grid cell and day are used as a base scaling coefficient (eq. (3)). Accurate emission estimation using the eq. (3) strongly depends on the simulation quality in COVID-19 pandemic. However, the study only shows a daily time series validation during whole years (I guess the domain-averaged concentrations. The paper did not mention). It is not enough for this study. The authors should provide an estimation in every grid-cell (or sites) near LNY-period.*

Thanks for the comment. Examples of emissions adjustment factors for NO2 and SO2 (January 15, 2020) were already provided in the supplementary materials (Figures S4 and S5). Evaluations for each monitor are provided in Figures S7-S9. We also modified the caption of Figure 4 to clarify the number of monitoring sites used.

*5. The study also introduces another coefficient β to furtherly adjust the ratio between observed and simulated pollutant concentrations. However, the major flow is that the study determines the β using two simulated concentration (adj1 and base in eq. (5)). Here the β in eq. (5) reflects the scaling relationship between simulation "adj1" and "base" but not the relationship between a simulation and an observation. In another word, the β in eq (5) should not be the one in eq. (3). Therefore, the introducing process of β in eq. (5) "as Equation (3) can be written (line 152)" are not reasonable.*

Thanks for the comment. We do not agree that β is mispresented in the equations. We double checked equations and confirm that β is properly presented.

By definition, $1/\beta$ is the sensitivity of concentrations due to perturbed emissions. One can calculate β with any model simulations with perturbed emissions. Here, we provide step-by-step descriptions for the equations.

From the manuscript, Eq. (3) is shown as

$$\frac{E_{adj}}{E_{mod}} = \beta \cdot \frac{C_{obs}}{C_{mod}}$$

Here, $C_{obs}$ is the concentration when emissions, $E_{adj}$, is applied to real world, or to model ($E_{adj1}$ and $C_{adj1}$). Therefore, the relationships, $E_{adj} : C_{obs} = E_{adj1} : C_{adj1} = E_{adj2} : C_{adj2}$, stand , and β is defined to represent this relationship.

For the first simulation, we applied $E_{adj1}$ to model

$$E_{adj1} = \frac{C_{obs}}{C_{base}} \cdot E_{base}$$

Actually, the choice of the perturbed emissions, $E_{adj1}$, is arbitrary, and we chose a case of β =1 for the simulation. (Again, you can calculate β for any model simulation. Lamsal et al. used a simple 30% reduction of emissions in their study.)

However, in the real world β ≠1, so the modeled concentrations, $C_{adj1}$, is not equal to $C_{obs}$. For the first simulation, Eq. (3) can be rewritten as

$$\frac{E_{adj1}}{E_{base}} = \beta \cdot \frac{C_{adj1}}{C_{base}}$$

By applying $E_{adj1}$, we can calculate β.

$$\beta = C_{obs}/C_{adj1}$$

At the second simulation, Eq. (3) can be rewritten as

$$\frac{E_{adj2}}{E_{base}} = \beta \cdot \frac{C_{adj2}}{C_{base}}$$

Since we need model concentrations is equal to observations ($C_{adj2} = C_{obs}$), and β was obtained from the first step, we can obtain the required emissions adjustment, $E_{adj2}$.

$$\frac{E_{adj2}}{E_{base}} = \beta \cdot \frac{C_{adj2}}{C_{base}} = \beta \cdot \frac{C_{obs}}{C_{base}} = \frac{C_{obs}}{C_{adj1}} \cdot \frac{C_{obs}}{C_{base}}$$

Therefore, the emission adjustment required for the second run (adj2) is

$$E_{adj2} = \frac{C_{obs}}{C_{adj1}} \cdot \frac{C_{obs}}{C_{base}} \cdot E_{base}$$

Please, think about the physical meaning of β. After one model simulation, if your model concentration, $C_{adj1}$, is still lower than observations, $C_{obs}$, you need to increase the model emissions to produce higher concentrations. β is the coefficient to provide that information.

*6. The study used the pollutant observation in the COVID-19 pandemic to validate the emission adjustment method (line 169). But considering the emission adjustment process is determined by the pollutant observations in the same period and grid-cells and β almost equal to 1, it is not a meaningful validation for the method.*

Thanks for the comment. We do understand reviewer's concern about potential self-validation. However, the key feature in this study is the validation of $PM_{2.5}$ concentration. We used observation-based SO2 and NO2 emissions adjustment. PM2.5 is totally independent. By updating SO2 and NOx emissions, the model was able to reproduce the changes of PM2.5 successfully. This is a strong

evidence showing that the top-down emissions adjustment method used this study is valid. Please, also see the general response 3.

---

## Author Response (AR1)

Authors' responses to reviewer comments

**"Quantitative assessment of changes in surface particulate matter concentrations over China during the COVID-19 pandemic and their implications for Chinese economic activity" by Kim et al.**

We thank the three reviewers and the editor for their productive comments, which enabled us to improve our manuscript. We provide below both general and point-by-point responses to the reviewers' comments.

**General response**

First, we present three major points in response to the reviewers' comments.

1.  Two independent analyses

In this study, we provide two analyses, which while independent suggest a consistent conclusion at the end. The time series analysis (Section 4.1) uses a bottom-up emissions inventory, and the top-down emissions adjustment experiment (Section 4.2) uses a top-down emissions inventory. While both analyses reach the same conclusion (i.e., ~30% missing of $PM_{2.5}$ emissions, potentially agricultural emissions), we never mixed the results, ensuring that the two analyses were conducted independently.

2.  Significance of analysis

In terms of regional air quality, the COVID-19 pandemic is a rare opportunity for a large-scale natural experiment for emissions control. This study suggests a novel method to update near real-time $NO_x$ and $SO_2$ emissions, and it demonstrates that this method works efficiently by comparing actual observations.

In terms of the COVID-19 pandemic, this study demonstrated that the change of economic sectors can be estimated based on pollutant concentrations. Several important conclusions were suggested, including the following:

(1) In order to assess quantitatively the changes in emissions, we need to isolate confounding factors from meteorological, emissions, and socioeconomic (e.g., Lunar New Year (LNY)) factors.
(2) There are different recovery speeds for different economic sectors, so changes in air quality and in emissions from COVID-19 lockdown cannot be generalized from examination of just one pollutant.
(3) There are potential missing emissions precursors for $PM_{2.5}$. This study suggests the potential lack of agricultural activity which, if correct, has crucial implications for agricultural production.

3.  Validation of top-down emission adjustments

We provide improved statistics by using updated $NO_x$ and $SO_2$ emissions. While we do understand the reviewers' concerns about potential self-validation, especially for $SO_2$ and $NO_2$ concentrations, the key feature in this study is the validation of $PM_{2.5}$ concentration. We used observation-based $SO_2$ and $NO_2$ emissions adjustment, and there was no adjustment in primary $PM_{2.5}$ emissions, meaning that $PM_{2.5}$ is totally independent. In this study, by updating $SO_2$ and $NO_x$ emissions, the model was able to reproduce the $PM_{2.5}$ changes successfully. This result confirms the importance of inorganic aerosols in current $PM_{2.5}$ concentrations. This evidence robustly demonstrates that the top-down emissions adjustment method used in this study is valid.

Below we provide point-by-point responses to the reviewers' comments.

**Point-by-point responses**

Reviewer #1

*L43: Surely residential, power generation and industry are all also major sources of NOx in China, such that transportation is not the majority source in many areas?*

Thank you for this observation. Transportation, power generation, and industry are three major sources of $NO_x$ emissions in Asia (Li et al., 2017). In China, the industrial sector was the most dominant, followed by power generation and transportation (in 2010) (Li et al., 2017). In the manuscript, we stated the role of the transportation sector since it adeptly represents characteristics of urban anthropogenic emissions. We also believe that the role of the transportation sector in $NO_x$ emissions will be prominent after long-term efforts to control emissions in industry and power generation, and with rapid growth of mega cities.

[Figure]

**Figure 4.** Emission distributions among sectors in Asia in 2010.

(Li et al., 2017)

| Regions | $SO_2$ | $NO_x$ | CO | NMVOC | $NH_3$ | $PM_{10}$ | $PM_{2.5}$ | BC | OC | $CO_2$ |
|---|---|---|---|---|---|---|---|---|---|---|
| **China** | **28 663** | **29 071** | **170 874** | **23 619** | **9804** | **16 615** | **12 200** | **1765** | **3386** | **10 124** |
| Power | 8166 | 9455 | 2077 | 255 | 0 | 1389 | 893 | 2 | 0 | 3245 |
| Industry | 16 775 | 11 218 | 71 276 | 14 461 | 239 | 9451 | 6061 | 575 | 530 | 4928 |
| Residential | 3489 | 1140 | 76 579 | 6349 | 450 | 5246 | 4737 | 908 | 2752 | 1266 |
| Transportation | 234 | 7257 | 20 942 | 2553 | 76 | 529 | 509 | 281 | 104 | 684 |
| Agriculture | | | | | 9040 | | | | | |

(Li et al., 2017)

*L48: Surely for the time duration being investigated here (weeks), meteorologically-driven variation in pollutant concentration is very important, and more important than natural inter-annual variations?*

Thank you for this comment, which has prompted us to modify the manuscript in order to clarify three major components: (1) natural variations, (2) emission control, and (3) sporadic socioeconomic events. Natural variations include impacts from short-term synoptic weather, interannual meteorological variations, and long-term climate change.

As a result of your useful observation, the revised manuscript now includes the following: "Three main components affect variations in pollutant concentrations: (1) natural variations (e.g., short-term synoptic weather, interannual meteorological variations, and long-term climate change), (2) long-term trends due to emissions control, and (3) sporadic socioeconomic events (Kim et al., 2017b)." (Line 48)

*L50: Yes, here the text does refer to meteorological variations so amend the phrasing of the equivalent point a couple of sentences earlier.*

We agree and have revised the manuscript accordingly. Please see the response above, and thank you for this observation.

*L67: State the time period or periods over which the 80% data availability criterion was applied.*

Thanks for the comment. For each year we used observational data during LNY-60 days to LNY+60 days, and selected sites with more than 80% data availability for each year (2017-2020, ± 60 days of LNY).

*Figure 1: Please increase the font size on the figure legend.*

We have updated Figure 1 accordingly.

Reviewer #2

*In the manuscript, the authors used 3-D chemical transport model along with other surface and remote measurements to study the change of surface concentrations in China due to the COVID-19 pandemic. One of their important strategies is to adjusting emission inventory which they claimed to be reasonable.*

Thank you for the comment. We did use two approaches and the emission adjustment is just one of them. This study offers two independent analyses that suggest the same conclusion. For more information, please refer to General Response 1.

*Still, I am not comfortable with their approach of adjusting emission inventory. I suggest the authors should provide more detailed explanation on it and how they could validate it.*

We provide a strong validation for the top-down emissions adjustment method in the comparison of $PM_{2.5}$ concentrations. Please refer to General Response 3.

*One more thing that concerns me is that it seems like that the manuscript does not contain significant scientific finding(s). It shows just the trend and some implications. Please provide why the results shown in the manuscript are important.*

Please refer to General Response 2 for the significance of the study.

Reviewer #3

*1. I do know why the authors named the paper as "Quantitative assessment of changes in surface particulate matter concentrations...", since the study mainly talks about the variations in emissions using model results and an emission-adjustment method. The comparison on surface particulate matter concentrations between the period and previous years has been showed by other studies or reports as the study mentioned. And we scarcely need a model assessment when we have the pollutant observations, considering the model results were adjusted by observations in the manuscript.*

To clarify, I assume that by "I do know" you meant "I don't know."

In a regional air quality problem, especially on the effects of anthropogenic emissions, the main cause and effect are the relationship between human activity, anthropogenic emissions, and pollutant concentrations. The bottom line is that we infer the extent of emissions change and related economic activities based on actual measurements of pollutant concentrations.

Since the concentration-emissions relationship depends strongly on the meteorological condition, one should consider the variations of meteorology and chemical reactions in order to claim the 'quantitative assessment' of emissions change or human activity. A simple comparison of concentration time series (as demonstrated in many previous studies) cannot do that.

We claim that our study is a quantitative assessment because we removed confounding factors by combining the information available from the chemistry transport model and also by applying data processing skills to remove variations from weekly, yearly, and LNY effects.

Please note, this study offers two independent analyses, both of which suggest a consistent result. We did not mix the analyses as the reviewer may think, but rather ensured that they were kept independent. For more information, please refer to General Response 1.

*2. The study claimed that "Meteorological influences were reduced by combining surface data with output from a three-dimensional chemistry model to calculate estimated emissions" (Section 3.1). I do not understand why combining the pollutant observations with model simulation can reduce the meteorological effects. The concentration time series, no matter from observation or from model, would be varied with the simulated/realistic meteorology. And, the adjusted emissions computed using eq. (1) or (6) should change with varied pollutant observations following meteorology.*

Thanks for this comment. The basic concept of the time series analysis in this study is that we need to retrieve emissions information out of measured concentrations. Given the same amount of emissions, the actual measured pollutant concentration would be significantly different based on the meteorological condition, especially in the formation of secondary pollutants. We combine 'observed concentrations' and 'the concentration-emission relation' from the model to estimate the emissions variations in the real world.

Please also note that we did not use the adjusted emission simulation for time series analysis. They are two separate analyses (please see General Response 1).

*3. Again, I am afraid that the "less sensitive of emission to meteorological variations" (line 119 and Fig. 2) would be not related to the combination process but due to the smoothing process with 7-day running average. If there is no any smoothing process, I believe the estimated emission time series would variation more sharply than Figure2 shown. Hence, the combination based on linear ratio of concentrations could not remove the meteorological influences on emission estimation.*

While we thank you for this comment, we respectfully disagree that the seven-day running averaging process is the only dominant data processing step. Changes in time series analyses after the seven-day moving average and after excluding meteorological impact were already demonstrated in Figure 7. As

such, we feel that it has already been made clear that both data processing procedures are important to capitalize the impact of the pandemic.

The seven-day moving average process was applied to remove unfair comparison by comparing different days of the week. Since we applied an alignment to center the LNY, daily time series comparisons were performed for different weekdays in different years because LNY days were assigned to different weekdays (please see table below). As the weekly variation of anthropogenic emissions is a dominant feature in the study of such emissions, we removed its noise by applying the seven-day moving average. If we simply compare time series without removing weekly variation, we compare signals from different weekdays. Since anthropogenic emissions have prominent weekly variations, this can result in unfair comparison. We do not conclude emissions changes by comparing emissions on Wednesday with emissions on Sunday, as it is not a fair comparison.

| Year | Lunar New Year | Weekday |
|------|----------------|---------|
| 2017 | January 28 | Saturday |
| 2018 | February 16 | Friday |
| 2019 | February 5 | Tuesday |
| 2020 | January 25 | Saturday |

*4. Top-down emission adjustments is the kernel of this study (Section 3.2). Here ratio between observed and simulated pollutant concentrations in every grid cell and day are used as a base scaling coefficient (eq. (3)). Accurate emission estimation using the eq. (3) strongly depends on the simulation quality in COVID-19 pandemic. However, the study only shows a daily time series validation during whole years (I guess the domain-averaged concentrations. The paper did not mention). It is not enough for this study. The authors should provide an estimation in every grid-cell (or sites) near LNY-period.*

We thank you for this point. Adjustment factors were calculated for each Chinese prefecture, and we respectfully refer you to BAE2020 for detailed technical data processing procedures. Examples of emissions adjustment factors for $NO_2$ and $SO_2$ (January 15, 2020) were already provided in the supplementary materials (Figures S4 and S5). Evaluations for each monitor are provided in Figures S7-S9. We also modified the caption of Figure 4 to clarify the number of monitoring sites used.

*5. The study also introduces another coefficient β to furtherly adjust the ratio between observed and simulated pollutant concentrations. However, the major flow is that the study determines the β using two simulated concentration (adj1 and base in eq. (5)). Here the β in eq. (5) reflects the scaling relationship between simulation "adj1" and "base" but not the relationship between a simulation and an observation. In another word, the β in eq (5) should not be the one in eq. (3). Therefore, the introducing process of β in eq. (5) "as Equation (3) can be written (line 152)" are not reasonable.*

While we thank you for this observation, respectfully we do not agree that β is mispresented in the equations. We double checked our equations and can confirm that β is properly presented.

By definition, 1/β is the sensitivity of concentrations due to perturbed emissions, and one can calculate β with any model simulations with perturbed emissions. Here, we provide step-by-step descriptions for the equations.

From the manuscript, Eq. (3) is shown as

$$\frac{E_{adj}}{E_{mod}} = \beta \cdot \frac{C_{obs}}{C_{mod}}$$

Here, $C_{obs}$ is the concentration when emissions, $E_{adj}$, are applied to the real world or to a model ($E_{adj1}$ and $C_{adj1}$). Therefore, the relationships, $E_{adj} : C_{obs} = E_{adj1} : C_{adj1} = E_{adj2} : C_{adj2}$, stand and β is defined to represent this relationship.

For the first simulation, we applied $E_{adj1}$ to our model

$$E_{adj1} = \frac{C_{obs}}{C_{base}} \cdot E_{base}$$

Actually, the choice of the perturbed emissions, $E_{adj1}$, is arbitrary, and we chose a case of β = 1 for the simulation. Again, you can calculate β for any model simulation. For example, Lamsal et al. used a simple 30% reduction of emissions in their study.

However, in the real world β ≠ 1, so the modeled concentrations, $C_{adj1}$, are not equal to $C_{obs}$. For the first simulation, Eq. (3) can be rewritten as

$$\frac{E_{adj1}}{E_{base}} = \beta \cdot \frac{C_{adj1}}{C_{base}}$$

By applying $E_{adj1}$, we can calculate β as follows:

$$\beta = C_{obs}/C_{adj1}$$

In the second simulation, Eq. (3) can be rewritten as

$$\frac{E_{adj2}}{E_{base}} = \beta \cdot \frac{C_{adj2}}{C_{base}}$$

Since we need model concentrations to equal observations ($C_{adj2} = C_{obs}$), and β was obtained from the first step, we can obtain the required emissions adjustment, $E_{adj2}$.

$$\frac{E_{adj2}}{E_{base}} = \beta \cdot \frac{C_{adj2}}{C_{base}} = \beta \cdot \frac{C_{obs}}{C_{base}} = \frac{C_{obs}}{C_{adj1}} \cdot \frac{C_{obs}}{C_{base}}$$

Therefore, the emission adjustment required for the second run (adj2) is

$$E_{adj2} = \frac{C_{obs}}{C_{adj1}} \cdot \frac{C_{obs}}{C_{base}} \cdot E_{base}$$

We hope this assuages your concerns, especially given the physical meaning of β. After one model simulation, if your model concentration, $C_{adj1}$, is still lower than observations, $C_{obs}$, you need to increase the model emissions to produce higher concentrations. β is the coefficient to provide that information.

*6. The study used the pollutant observation in the COVID-19 pandemic to validate the emission adjustment method (line 169). But considering the emission adjustment process is determined by the pollutant observations in the same period and grid-cells and β almost equal to 1, it is not a meaningful validation for the method.*

We understand the reviewer's concern about potential self-validation, and thank you for this comment. However, the key feature in this study is the validation of $PM_{2.5}$ concentration. We used observation-based $SO_2$ and $NO_2$ emissions adjustment. $PM_{2.5}$ is totally independent. By updating $SO_2$ and $NO_x$ emissions, the model was able to reproduce the changes of $PM_{2.5}$ successfully. This is strong evidence showing that the top-down emissions adjustment method used in this study is valid. For more information, please see General Response 3.

---

## Referee Report (RR1)

The author has taken some efforts to improve the manuscript. However, I still have more concerns about the manuscript and the authors' responses.

1. To be honest, I am not clear about the significance of this study. If we want to present the decreased human activities in the COVID-19 pandemic using pollutant concentrations, an analysis of observed $NO_2$, $SO_2$, and $PM_{2.5}$ is enough. Why we need the simulated $NO_2$, $SO_2$, and $PM_{2.5}$ using the modified emissions that are adjusted by observed $NO_2$, $SO_2$?

2. About the title. The main purpose of this work is to infer the changes in human activities in the COVID-19 pandemic, which can be directly reflected by changes in emissions over China. Please note that the change in emissions is more important than changes in PM2.5 concentrations in the context of the manuscript. For example, most sentences in the summary section are about the changes in emissions other than PM2.5 concentrations. Hence, please revise the title of the manuscript to reflect the changes in emissions.

3. I do understand meteorology should be excluded when retrieving emissions out of measured concentrations. The authors should clarify why "meteorological influences were reduced by combining surface data with output from a three-dimensional chemistry model to calculate estimated emissions" in the manuscript.

4. I believe several days smooth is important for the method. Otherwise, the adjusted emissions will vary very sharply. The explanation of the seven-day smoothing process is not convincing. As I know, there is a long period of LNY holidays every year in China, which should robustly impact the anthropogenic emissions and pollutant concentrations near the LNY-period. Hence, there may be no clear weekly variations in China in the period. At least, the authors should compute the significance of the weekly variations in that period to support the validity of the seven-day smoothing process.

5. Fig. S7-9 just provides a spatial estimation of the model performance. It is not very important for the study. Following Fig 4, the authors should provide time-series estimations in every grid-cell (or sites) near LNY-period. For example, the spatial distribution of temporal correlation coefficients or temporal RMSEs is needed.

6. The Authors repeated the equations in detail on their method. But respectfully I do not very agree with this explanation. The major flaw is that the β is set to be a fixed coefficient (i.e., linear relationship) by default for any model simulations. I will illustrate that through four aspects.

1) In the response, the authors use the equation (page 6)
$$\frac{E_{adj}}{E_{mod}} = \beta \cdot \frac{C_{obs}}{C_{mod}} \qquad (A)$$

which are applied to the real world or to a model ($E_{adj1}$ and $C_{adj1}$) (the first sentence of page 7) and derive the relationships, $E_{adj}: C_{obs} = E_{adj1}: C_{adj1} = E_{adj2}: C_{adj2}$ (the second sentence on page 7). Please note that the relationships stand just when $\beta$ is unchanged for *adj*, *adj1*, and *adj2* according to eq. (*A*). But, at least in *adj1* and *adj2* simulations, the authors use $\beta = 1$ and $\beta \neq 1$ respectively. On the contrary, if authors believed such different $\beta$ setting in *adj1* and *adj2* simulations are both reasonable, the relationships $E_{adj}: C_{obs} = E_{adj1}: C_{adj1} = E_{adj2}: C_{adj2}$ cannot stand.

2) In experiment *adj1*, the authors chose arbitrary $\beta = 1$ for the simulation. As a result, the adjusted emission $E_{adj1}$ and simulated $C_{adj1}$ are arbitrary. $C_{adj1}$ is not equal to $C_{obs}$, and $E_{adj1}$ is not the emissions corresponding to $C_{obs}$. In this case, why the eq. (*A*) still stands for *adj1*?

3) This method implies that the value of $\beta$ is unchanged no matter what $\beta$ they chose in *adj1* (here the authors chose arbitrary $\beta = 1$). I am afraid $\beta$ would change when choosing a very large (10 as an example) or very small (0.1 as an example) $\beta$ in *adj1* because the large scaling in emission will cause non-linear responses to pollutant concentrations. Please show the readers that the spatial distribution of $\beta$ is unchanged when using different $\beta$ (for example, 0.1, 1 and 10) in *adj1*.

4) Again, linear change in emissions does not cause a linear change in concentrations, considering many non-linear impacts of chemical reaction, deposition processes and meteorology. Hence, simulations with different emission amounts should have a different relationship between emission and concentrations. In another word, for *adj1* (without regard to the point (2))

$$\frac{E_{adj1}}{E_{base}} = \beta_1 \cdot \frac{C_{adj1}}{C_{base}}$$

and for *adj2*

$$\frac{E_{adj2}}{E_{base}} = \beta_2 \cdot \frac{C_{adj2}}{C_{base}}$$

Apparently, the manuscript implied $\beta_1 = \beta_2$ without any explanation. Hence, please show the readers why the $\beta$ derived from *adj1* can be directly applied to *adj2*.

7. Once reducing the emissions in $SO_2$ and NOx (two critical precursors for $PM_{2.5}$ considering the abundant $NH_3$ over China) in the model according to observed $NO_2$ and $SO_2$, the $PM_{2.5}$ concentrations generally approaches to observations. Hence, it is not surprised to get good $NO_2$, $SO_2$ and $PM_{2.5}$ simulations. From this point, $PM_{2.5}$ is not "totally independent". Please add some discussion for the validation.

---

## Author Response (AR2)

Authors' responses to the reviewers' comments

**"Quantitative assessment of changes in surface particulate matter concentrations and precursor emissions over China during the COVID-19 pandemic and their implications for Chinese economic activity" by Kim et al.**

We again thank the three reviewers and the editor for their productive comments. We notice there are strong negative comments from one reviewer (#3). After careful deliberation, we believe that there are two critical misunderstandings about our study that may have caused the negative review. We accept that it is our responsibility to provide clear descriptions on the methodology in the manuscript and we apologize for its shortcomings. In the revised manuscript, we have improved the manuscript to better describe the methodology and analyses.

We will provide further explicit details below, but to start, we would like to summarize three key discrepancies between claims in Reviewer #3's comments and what was actually done in the study.

1. We did not apply the adjusted emissions for the time series analysis, as claimed by Reviewer #3. In this study, we demonstrated several complementary analyses using bottom-up emissions inventories and top-down adjusted emissions estimates. For the first analysis -- the time series analysis -- we utilized the model simulations together with a fixed emissions inventory because model simulations with fixed emissions can provide the pure impact of meteorological variations.

2. We did not use a fixed $\beta$ value (i.e., the sensitivity of the concentration to the emissions change) for the simulations, as claimed by Reviewer #3. We calculated individual $\beta$ values for each Chinese prefecture, for each day, and for every chemical component (i.e., for $NO_2$ and $SO_2$ separately).

3. Updating $SO_2$ or $NO_x$ emissions based on the observations, especially from satellites, is a widely used practice in the applications of top-down emissions in the regional air quality modeling community. While we did this approach more carefully, by calculating specific emissions-to-concentrations sensitivities, the fundamental of this approach is straightforward and common, which we do not believe to be unconventional or contentious.

In the following, we address Reviewer #3's comments (shown in boxes) in detail and describe how the manuscript was changed to address these comments.

> The author has taken some efforts to improve the manuscript. However, I still have more concerns about the manuscript and the authors' responses.
>
> 1. To be honest, I am not clear about the significance of this study. If we want to present the decreased human activities in the COVID-19 pandemic using pollutant concentrations, an analysis of observed $NO_2$, $SO_2$, and $PM_{2.5}$ is enough. Why we need the simulated $NO_2$, $SO_2$, and $PM_{2.5}$ using the modified emissions that are adjusted by observed $NO_2$, $SO_2$?

This manuscript presents two experiments using model simulations with fixed emissions (i.e., normal bottom-up emissions inventory) and adjusted emissions (i.e., top-down emissions inventory). The first analysis (time-series analysis) in Section 4.1 using observations coupled with fixed-inventory model simulations provides much more useful and accurate estimates of emissions than simply using observations alone. If only observations were used, as the Reviewer suggests, then variations in observations due to meteorological variations would be misinterpreted as emissions changes. Our analysis has attempted to identify the actual emissions changes, based on observations, by removing the variations in observations caused simply by meteorological variations.

We have made the following changes in the manuscript to attempt to clarify this issue:

- Lines 103-106 were revised to clarify that we used fixed emission inventory for the time series analysis (Section 4.1) and Section 4.2 describes the emissions adjustment experiment. Section 4.1 and 4.2 are independent analyses.

  "This section describes the following aspects of the analysis: (1) data-processing procedures for analyzing the time series, (2) emissions-adjustment procedures to update $SO_2$ and $NO_x$ emissions to near real-time, and (3) brute-force modeling procedures to estimate Chinese emissions by sector. It should be noted that the time series analysis (discussed in Section 4.1) utilizes fixed emissions inventory (i.e. bottom-up emissions inventory) and the emission adjustment experiment (Section 4.2) utilizes observation-based top-down emissions. Sectoral emissions estimations method is for Section 4.3."

- Lines 108-113 – a new sentence (in blue) has been added to clarify this issue:

  "Four types of variation (meteorological, weekly, yearly, and the Chinese spring festival) were reduced or accounted for in the surface observations, as follows. Meteorological influences were reduced by combining surface data with output from a three-dimensional chemistry model to calculate estimated emissions. Since the model simulations with fixed emissions inventory respond to the variations of meteorological conditions, we can infer the relationship between emissions and ambient pollutant concentrations under a specific weather condition. By applying this relationship, we convert the changes of observed concentrations into the changes of emissions."

> 2. About the title. The main purpose of this work is to infer the changes in human activities in the COVID-19 pandemic, which can be directly reflected by changes in emissions over China. Please note that the change in emissions is more important than changes in PM2.5 concentrations in the context of the manuscript. For example, most sentences in the summary section are about the changes in emissions other than PM2.5 concentrations. Hence, please revise the title of the manuscript to reflect the changes in emissions.

- We have revised the title. The new title -- with changes in blue text – is:

  "Quantitative assessment of changes in surface particulate matter concentrations and precursor emissions over China during the COVID-19 pandemic and their implications for Chinese economic activity"

> 3. I do understand meteorology should be excluded when retrieving emissions out of measured concentrations. The authors should clarify why "meteorological influences were reduced by combining surface data with output from a three-dimensional chemistry model to calculate estimated emissions" in the manuscript.

This question addresses one of the fundamental aspects of the manuscript, and we have tried in this overall response and in the changes made to the manuscript to make it more understandable.

We have made the following changes in the manuscript to attempt to clarify this issue:

- Lines 108-120 have been expanded with new explanations (in blue) (note that the first change in the paragraph below was already noted in response to comment #1 above):

  "Four types of variation (meteorological, weekly, yearly, and the Chinese spring festival) were reduced or accounted for in the surface observations, as follows. Meteorological influences were reduced by combining surface data with output from a three-dimensional chemistry model to calculate estimated emissions. Since the model simulations with fixed emissions inventory respond to the variations of meteorological conditions, we can infer the relationship between emissions and ambient pollutant concentrations under a specific weather condition. By applying this relationship, we convert the changes of observed concentrations into the changes of emissions. Weekly variations, a unique feature of anthropogenic emissions, were removed by using a seven-day moving average. The impact of the Chinese spring festival, the biggest traditional holiday celebrating Lunar New Year (LNY), was normalized by rearranging the time series to center on the LNY in each solar year. The LNY alignment was necessary to account for the irregular happening of the LNY dates. Seven-day moving average filtering was also required to avoid unfair comparisons between different weekdays after the LNY alignment. Otherwise, we may compare different weekdays for different year (e.g. 2020 LNY on January 25, Saturday and 2019 LNY is February 5, Tuesday). **Figure S4** shows that the seven-day moving average filter smooths but does not significantly change the time-series results. Finally, yearly emission variations were removed by setting a base period (-60 to -10 days before LNY) and calculating relative changes from the average of the base period."

4. I believe several days smooth is important for the method. Otherwise, the adjusted emissions will vary very sharply. The explanation of the seven-day smoothing process is not convincing. As I know, there is a long period of LNY holidays every year in China, which should robustly impact the anthropogenic emissions and pollutant concentrations near the LNY-period. Hence, there may be no clear weekly variations in China in the period. At least, the authors should compute the significance of the weekly variations in that period to support the validity of the seven-day smoothing process.

As we explained in the previous response, the seven-day moving average process was applied to remove unfair comparisons by comparing different days of the week. Since we applied an alignment to center the LNY, daily time series comparisons were performed for different weekdays in different years because LNY days were assigned to different weekdays.

- The text addition to the manuscript shown in the response to comment #3 above addresses this issue.

- We have also added a new figure in the Supplementary Material (Figure S4) to show the time-series analysis with and without seven-day moving-average data processing (see below). While the time-series results without seven-day moving-average processing is a little noisier due to the unfair comparison between different weekdays, we do not see significant differences between the two plots. Therefore, we believe that the conclusions drawn from this portion of the analysis stand irrespective of the use of seven-day moving-average data processing in the analysis.

[Figure]

Figure S4. Comparison of the original time series by removing meteorological, weekly, yearly and the LNY signals (left) and the one without seven-day moving average (right). The seven-day moving average filtering is required to avoid unfair comparisons between different weekdays after the LNY alignment.

5. Fig. S7-9 just provides a spatial estimation of the model performance. It is not very important for the study. Following Fig 4, the authors should provide time-series estimations in every grid-cell (or sites) near LNY-period. For example, the spatial distribution of temporal correlation coefficients or temporal RMSEs is needed.

Here, "time series estimations in every grid-cell (or site) near the LNY period" means more than 1500 time series plots. We do not believe that inclusion of these individual-site plots is practical or useful for the manuscript. We strongly believe that the temporal summary of this information (Figure 4) and spatial summary of this information (Figures S7-9) are the most useful ways to present this information.

- We have included an example of a time series plot for one site as a new figure, Figure S11, for Kuang, Handan (lon=114.504, lat=36.5776, id=1049A).

- We have also generated time series plots at all individual monitoring sites per the reviewer's request, and plots for all 1570 sites are available at an external link (https://www.dropbox.com/s/e8czqza66jpcxz1/out-ts-all.tgz?dl=0).

[Figure]

Figure S11. Time series and scatter plots of observed and modeled surface concentrations of SO2, NO2, and PM2.5 from the Kuang, Handan monitoring site (lon=114.504, lat=36.5776, id=1049A).

- We have also provided a new figure (Figure S12) showing the spatial distribution of the RMSE for the *base* and *adj2* runs during February and March 2020. These are consistent with the bias spatial plots that had already been provided in the supplementary information.

[Figure]

Figure S12. Spatial distributions RMSE for the base (left) and the adj2 run (right). RMSEs were calculated from daily mean concentrations during February and March 2020 for each monitor.

6. The Authors repeated the equations in detail on their method. But respectfully I do
not very agree with this explanation. The major flaw is that the β is set to be a fixed
coefficient (i.e., linear relationship) by default for any model simulations. I will
illustrate that through four aspects.

The reviewer's claim that we used a fixed β for the study is not true. We calculated the β values for all locations and times. We also calculated the β values for $NO_x$ and $SO_2$ separately. However, in most cases, the β values are slightly over one, confirming that those emissions are mostly primary.

We have included an extensive new discussion on the emissions-to-concentration sensitivities (i.e., β values) (Section 4.4.2) to clarify this issue. We have investigated the spatial, temporal, and chemical characteristics of the β values, including a new figure (Figure 8) and concluded that they are mostly consistent for a specific location and chemical component.

- The following text has been added (lines 369-379), and Figure 8 has been added:

[revised manuscript text omitted]

The β values are calculated as follows. In the real world, the sensitivity of concentration to changes in emissions is not unique or spatially homogeneous (i.e., $\beta \neq 1$), especially for $NO_x$ emissions and $NO_2$ concentrations. β values for specific location and time can be calculated if we have two model simulations with different emissions applied. Previous studies have calculated β values for a model by using changes in concentration caused by a certain amount of perturbed emissions (e.g., Lamsal et al., 2011 used a 15% emissions pertubation).

To obtain more realistic β values, we have conducted two model simulations, *base* and *adj1* runs. First, the *base* model simulation was conducted using normal emissions inventory, CREATE, we have introduced previously. The second simulation, *adj1* run, was conducted using perturbed emissions to estimate how the model responds according to the change of emissions. We adjusted emissions according to the ratio between observed and modelled surface concentrations, so we can reproduce more realistic chemical environment.

From these two simulations, the *base* and *adj1* runs, we calculate the emissions-to-concentration sensitivity, β values, in specific spatial and temporal scale – for each Chinses prefecture daily. β values are calculated as,

$$\beta_{p,t} = \frac{[E_{adj1}/E_{base}]_{p,t}}{[C_{adj1}/C_{base}]_{p,t}} \tag{4}$$

where *p* and *t* stand for indices of Chinese prefectures and specific dates. Using calculated β values for each prefecture and date, we finally obtain the adjusted emissions for the second and final simulations, *adj2* run.

$$[E_{adj2}]_{p,t} = \beta_{p,t} \cdot \left[\frac{C_{obs}}{C_{base}} \cdot E_{base}\right]_{p,t} \tag{5}$$

We further discuss the characteristics of the emissions-to-concentration sensitivity in Section 4.4.2."
* * *
1) In the response, the authors use the equation (page 6)

$$\frac{E_{adj}}{E_{mod}} = \beta \cdot \frac{C_{obs}}{C_{mod}} \tag{A}$$
* * *
which are applied to the real world or to a model ($E_{adj1}$ and $C_{adj1}$) (the first sentence of page 7) and derive the relationships, $E_{adj}: C_{obs} = E_{adj1}: C_{adj1} = E_{adj2}: C_{adj2}$ (the second sentence on page 7). Please note that the relationships stand just when β is unchanged for *adj*, *adj1*, and *adj2* according to eq. (A). But, at least in *adj1* and *adj2* simulations, the authors use $\beta = 1$ and $\beta \neq 1$ respectively. On the contrary, if authors believed such different β setting in *adj1* and *adj2* simulations are both reasonable, the relationships $E_{adj}: C_{obs} = E_{adj1}: C_{adj1} = E_{adj2}: C_{adj2}$ cannot stand.

As noted and clarified above, we calculated the β values for all locations, times, and chemical components separately. And as noted above, in the revised manuscript, the β values for the same location, time, and chemical component are mostly consistent.
* * *
2) In experiment *adj1*, the authors chose arbitrary $\beta = 1$ for the simulation. As a result, the adjusted emission $E_{adj1}$ and simulated $C_{adj1}$ are arbitrary. $C_{adj1}$ is not equal to $C_{obs}$, and $E_{adj1}$ is not the emissions corresponding to $C_{obs}$. In this case, why the eq. (A) still stands for *adj1*?

Again, in this study, we calculated the β values for all locations, times, and chemical components separately.

3) This method implies that the value of $\beta$ is unchanged no matter what $\beta$ they chose in *adj1* (here the authors chose arbitrary $\beta = 1$). I am afraid $\beta$ would change when choosing a very large (10 as an example) or very small (0.1 as an example) $\beta$ in *adj1* because the large scaling in emission will cause non-linear responses to pollutant concentrations. Please show the readers that the spatial distribution of $\beta$ is unchanged when using different $\beta$ (for example, 0.1, 1 and 10) in *adj1*.

For the *adj1* run, we have updated the emissions according to the observation-to-model concentrations of the *base* run. Then, we have explained that this adjustment is consistent with β=1. In this study, we calculate the β values out of the two model simulations; we do not use an arbitrary β value.

- We have included a new section 4.4.2 that describes the *adj1* run in much greater detail.

"As stated in the methodology section, we further discuss here the emissions-to-concentration sensitivities (i.e. β). The β values can be calculated using any two model simulations based on different emissions inputs, by comparing the change in emissions with the change in simulated concentrations. Furthermore, if we specifically change the emissions according to the ratio of observations and the base model simulation, we further simplify the emissions scaling factor as follows.

For this simulation, adj1, if we apply the adjusted emissions using the ratio of the observed and modeled concentrations, the adjusted emissions for the adj1 run, $E_{adj1}$, are

$$E_{adj1} = \frac{C_{obs}}{C_{base}} \cdot E_{base} \qquad (6)$$

If we apply this to Eq. (4), we can obtain

$$\beta = \frac{E_{adj1}/E_{base}}{C_{adj1}/C_{base}} = \frac{C_{obs}/C_{base}}{C_{adj1}/C_{base}} = \frac{C_{obs}}{C_{adj1}} \qquad (7)$$

Therefore, the emission adjustment factors in the next simulation (adj2) can be found using Eq. (5):

$$E_{adj2} = \beta \cdot \frac{C_{obs}}{C_{base}} \cdot E_{base} = \left[ \frac{C_{obs}}{C_{adj1}} \cdot \frac{C_{obs}}{C_{base}} \right] \cdot E_{base} \qquad (8)$$

where adj2 indicates the second and final simulation for the top-down emissions adjustment method.

From here, the $\left[\frac{C_{obs}}{C_{adj1}}\right]$ term, or β, can be interpreted as an additional adjustment factor to the original adjustment factor in adj1, $\left[\frac{C_{obs}}{C_{base}}\right]$. If the emissions modification in adj1 results in the same percentage change in concentrations, $C_{obs}$ / $C_{adj1}$ = 1, we do not need the secondary adjustment. If the simulated concentration from adj1 is smaller (larger) than the observations, we need to increase (reduce) the amounts of emissions. This procedure was applied to create new 2020 emissions of both $SO_2$ and $NO_x$.

In most cases, the calculated β values are close to one (**Figure S4**), implying that the simple assumption β = 1 in BAE2020 remains effective. The β values for $NO_x$ emissions are slightly higher than those for $SO_2$ emissions over polluted areas (**Figure S5**), which implies that more secondary reactions are involved in tropospheric $NO_x$ chemistry.

Both enhancements to the top-down simulations—β values and the daily application of emission adjustment factors—clearly improved the model's performance, especially in the pre-LNY periods. While the monthly emissions adjustments failed to represent the rapid changes in $NO_2$ concentrations after January 25, 2020 (**Figure S6**), the daily adjustment method successfully modeled these changes (**Figure 4**). The general underestimation of $NO_2$ concentrations was corrected using the β values (**Figure 4**). The improved model performance was confirmed by comparing the spatial distributions and scatterplots before and after these adjustments (**Figures S7–S9**).

Understanding the characteristics of the β values in terms of their spatial distribution, temporal variation, and chemical difference is important for several reasons. In the emission update procedure in practice, we can apply the pre-calculated β values from the look-up table if the β values show general consistency according to their location, time, and chemical component. For the emission control policy, the β values provide valuable information on the efficiency of emissions control because they suggest how effectively pollutant concentrations can be removed given the amount of emissions control by the government."
* * *
4) Again, linear change in emissions does not cause a linear change in concentrations, considering many non-linear impacts of chemical reaction, deposition processes and meteorology. Hence, simulations with different emission amounts should have a different relationship between emission and concentrations. In another word, for *adj1* (without regard to the point (2))

$$\frac{E_{adj1}}{E_{base}} = \beta_1 \cdot \frac{C_{adj1}}{C_{base}}$$

and for *adj2*

$$\frac{E_{adj2}}{E_{base}} = \beta_2 \cdot \frac{C_{adj2}}{C_{base}}$$

Apparently, the manuscript implied $\beta_1 = \beta_2$ without any explanation. Hence, please show the readers why the β derived from *adj1* can be directly applied to *adj2*.
* * *
β is the emissions-to-concentration sensitivities. For the same time and location as well as for the same chemistry model, the β value should be identical for *adj1* and *adj2*.

7. Once reducing the emissions in $SO_2$ and NOx (two critical precursors for $PM_{2.5}$ considering the abundant $NH_3$ over China) in the model according to observed $NO_2$ and $SO_2$, the $PM_{2.5}$ concentrations generally approaches to observations. Hence, it is not surprised to get good $NO_2$, $SO_2$ and $PM_{2.5}$ simulations. From this point, $PM_{2.5}$ is not "totally independent". Please add some discussion for the validation.

We completely agree that PM2.5 is "chemically" related to precursor emissions. That is why we found much better model performance by updating the $NO_x$ and $SO_2$ emissions. This provides clear evidence to demonstrate the efficiency of our top-down emissions update methodology.

While the $NO_2$ and $SO_2$ emissions are primary inputs for the $NO_2$ and $SO_2$ concentrations, the $PM_{2.5}$ concentration is mostly controlled by secondary chemical reactions. $PM_{2.5}$ concentrations, especially inorganic components, are determined by the balance of nitrate-surface-ammonium formations. The term "totally independent" means that we did not adjust any primary $PM_{2.5}$ emissions. We only adjusted the $NO_x$ and $SO_2$ emissions, and those changes chemically improved the $PM_{2.5}$ simulations significantly, through complicated chemical reactions and balances, within the chemistry model.

We believe that this provides strong evidence that the top-down emission adjustment method worked "chemically".

We would also like to note that, as the editor suggested, emissions update training was already applied separately for $SO_2$ and $NO_x$. No primary $PM_{2.5}$ emissions were adjusted in the study. Therefore, $SO_2$ and $NO_x$ emissions were *trained* using observations, and $PM_{2.5}$ concentrations modeling performance was improved by chemical procedures, *validating* the top-down emission update approach.

The following text was added to discuss the validation:

> "To evaluate the emissions update approach, the key feature in this study is the validation of $PM_{2.5}$ concentration. We used observation-based $SO_2$ and $NO_2$ emissions adjustments and there was no adjustment in the primary $PM_{2.5}$ emissions, meaning that the improvement of $PM_{2.5}$ is achieved through chemical reactions and their balances. The surface concentrations of surface $PM_{2.5}$ concentrations, especially inorganic aerosols, are formed by secondary reactions, which are determined by the balance of chemical reactions for nitrate, sulfate, and ammonium. The performance of the $PM_{2.5}$ simulations provides strong evidence that the top-down emissions adjustment method used in this study is valid and successfully reproduces a realistic chemical environment."

---

## Author Response (AR3)

Authors' responses to the reviewers' comments

**"Quantitative assessment of changes in surface particulate matter concentrations and precursor emissions over China during the COVID-19 pandemic and their implications for Chinese economic activity" by Kim et al.**

We again thank the two reviewers and the editor for their productive comments. We provide below point-by-point responses to the reviewers' comments. Reviewers' comments are shown in italics.

Reviewer #1
*Major comments:*
*-Section 1: In the introduction section, the scientific significance drawn from this study should be clarified. On page 2, lines 54-57, previous studies in Chinese air quality are simply summed up. I would like to disagree this rough introduction for previous researches. What have been already known and what are remained subjects should be politely introduced here. This will reinforce the significance of this manuscript.*

We thank the reviewer for this comment. We have revised the manuscript to include previous works on COVID-19 study with their areas of investigations.

Line 54

"Although early studies have reported Chinese air quality during the period in question, in terms of surface observations and air quality indices (Bao and Zhang, 2020; Chauhan and Singh, 2020; He et al., 2020; Shi and Brasseur, 2020; Xu et al., 2020), satellite observations (Liu et al., 2020a, 2020b), atmospheric chemistry modeling (Kang et al., 2020; Li et al., 2020; Wang et al., 2020), emissions estimation via inverse modeling (Miyazaki et al., 2020; Zhang et al., 2020), secondary aerosol formation (Huang et al., 2020), and human activity and energy use (Wang and Su, 2020), it remains challenging to fully isolate the impact of the pandemic on the region's air quality."

*-Section 2.2: The satellite data of TROPOMI seems to be only noted in page 7, lines 239−240. Through the manuscript, I found the wording of "top-down" estimate. In my experiences, this wording is usually state the satellite-constrained data assimilation/inversion method. However, as far as I catch up from Section 3.2, the satellite data of TROPOMI is not used for emission adjustment by β-method. If the satellite data is not used in β-method, I like to avoid the wording of "top-down" throughout the manuscript and completely move Section 2.2 into supplemental material for well-ordered manuscript. Please consider this point in revision process.*

We thank the reviewer for the comment. Although the satellite observations have been widely used for top-down emissions estimation, due to their extensive spatial coverage, the term of "top-down" emission inventory is not limited to the use of satellite data. "Top-down" and "bottom-up" emissions inventory methods indicate two different emissions inventory construction methods, "observation-based" and "activity-based", respectively. The top-down method utilizes the total amount of emissions from observations (e.g. satellite, in situ and surface monitors) and the bottom-up method utilizes survey-based information and emissions factors to sum up all individual emissions activities. Top-down does not specifically indicate that the monitoring method is "physically looking down".

We offer the following examples in the literature and from presentations for use of the top-down and bottom-up emission inventory methodologies. We believe the top-down term is well-understood in the research community to comport with our usage, and so we would like to keep the term as it is in our manuscript.

- https://www.geiacenter.org/sites/default/files/site/community/geia-conferences/2015/presentations/topdown%20emissions%20analyses%20theme/Session%201/2.GEIA2015TopDown_Frost_18Nov15.pptx

- https://www.nrel.gov/news/program/2018/natural-gas-emissions-measure-top-down-or-bottom-up.html
- Cheewaphongphan et al. (2019), doi:10.3390/su11072054

*-Section 4.2: Because the original model simulation showed super overestimate for SO2, this study presented to update SO2 emissions based on β-method. I have two questions for this approach.*
*--From the updated SO2 emissions, the model performance for SO2 concentration have been dramatically improved. This is just based on the assumption that SO2 concentration is depend on SO2 emissions, and SO2 emissions is forced to be adjusted. As reported in the papers for emission inventories, the uncertainty on the estimation of SO2 emissions is relatively lower than other pollutants because the sources for SO2 is generally well-known (power plant and industry). Even though for the purpose of improving the model performance for SO2 concentration, I am wondering such re-calculated SO2 emissions is reliable data.*

We thank the reviewer for the comment. As the reviewer commented, the $SO_2$ emissions sources are relatively well characterized and have lower uncertainties. However, their temporal variation can be significant due to the change of economic condition or government emission control policy. Since reducing $SO_2$ emissions has been a high priority for the Chinese government, annual emissions have been reduced dramatically in recent years. Bottom-up emission inventories take a lot of time and resources to construct and so are easily outdated.

Figure R1 shows the time series of surface $SO_2$ concentration across China during 2015 to 2020. Annual mean concentrations decreased dramatically (i.e., 9.8 ppb (2015), 8.4 ppb (2016), 6.9 ppb (2017), 5.1 ppb (2018), 4.2 ppb (2019), 3.7 ppb (2020)). In the study, we used a 2016 emissions inventory, and, in 2020, annual mean $SO_2$ concentration was already less than the half of 2016 level. Although the emissions inventory used in the study is well developed through in situ measurements and is believed to be relatively accurate for 2016, the inventory does not reflect the subsequent changes in emissions. Until updated bottom-up inventories are available, the use of top-down emission update methodologies are very useful.

We have included this discussion in the manuscript.

Line 144

"Due to stringent emissions control policies by the Chinese government, Chinese anthropogenic emissions changed dramatically over recent years. For example, the annual mean surface $SO_2$ concentration across China was 8.4 ppb in 2016, but dropped to less than half of this level (3.7 ppb) in 2020."

[Figure]

*Figure R1 Daily average surface SO2 concentrations across China during 2015-2020.*

*--In spite of the large reduction of SO2 emissions through updated emissions, PM2.5 concentration did only show slight decline (e.g., Figure 4). SO42- would be produced most linearly according to precursor SO2, but why? Because SO42- is one of the dominant species on PM2.5, I simply expect much decline on PM2.5 due to the updated emissions. To support this discussion, the analysis of PM2.5 composition is required, but is there no data for it?*

Thanks for this comment.

Unfortunately, PM speciation data is not available for the year 2020. Instead, we analyzed variations of simulated PM components. In the simulation, nitrate is the most dominant component in $PM_{2.5}$ (51%), followed by ammonium (20%) and sulfate (14%). Based on this analysis, PM2.5 concentrations are not expected to be strongly influenced by sulfate concentrations or the change of $SO_2$ emissions.

Figure R2 demonstrates times series of PM-component changes between the base and the adjusted simulations. Under ammonia-rich chemical condition, the chemical balances in nitrate-sulfate-ammonium chemistry control the final concentrations of $PM_{2.5}$. From the simulation, the efficiency of conversion of $SO_2$ emissions to $SO_4$ aerosol is not great, likely due to low chemical reactivity during the wintertime. In addition, during the pre-pandemic period, the $PM_{2.5}$ reduction in sulfate is mostly canceled out by the increase of nitrate concentration. During the pandemic period, the change of nitrate concentration is a major driver for the total $PM_{2.5}$ concentrations. This can be an important message in emission reduction policy since PM pollutions can be efficiently controlled when both $SO_2$ and $NO_x$ emissions are controlled. Clearly, this emission control efficiency is an attractive topic to pursue in future studies. However, we believe that full investigation of this topic is beyond the scope of the current manuscript. Further investigations, including analysis like Figure R2, will be reported in future work.

We have revised the manuscript to include this discussion.

Line 362

"Formation efficiency of sulfate aerosols by updating $SO_2$ and $NO_x$ emission is also very interesting. From Figure 4, one may notice that the change of total $PM_{2.5}$ concentration is not prominent in the pre-pandemic period, even with strong reduction in $SO_2$ emissions. Modelled PM speciation components show that the reduced sulfate concentrations were cancelled out by the increased nitrate

concentrations, due to the balance of non-linear nitrate-sulfate-ammonium chemistry. Nitrate is the most dominant component of $PM_{2.5}$ during the wintertime (contributing ~50% while sulfate contributes 14%), and the sudden drop of $PM_{2.5}$ concentrations during the pandemic is mostly driven by the change of nitrate concentrations. This result implies an important message to emissions control policy, suggesting that both $SO_2$ and $NO_x$ emissions reductions will be required to achieve better emission reduction efficiency.

[Figure]

*Figure R2 Time series of PM speciation components change between the base and the adjusted simulations during the pandemic period. Changes of sulfate (ASO4), nitrate (ANO3), ammonium (ANH3) and PM2.5 concentrations are demonstrated. X marks indicate the sum of sulfate-nitrate-ammonium concentration changes.*

*-Section 4.3 and Figure 6: The approach to evaluate the sectoral contribution, the authors used BFM with 50% reduction. Due to the nonlinearity, the total sector contributions may not be matched to 100% in some cases, whereas the result showed exact 100%. Did the authors normalize the contribution? Please add the explanation to draw this result.*

Thanks for this comment. As the reviewer mentioned, fractional contributions were calculated compared to the sum of total contributions from five emission sectors. We have clarified it in the manuscript.

Line 188

"Fractional contributions of each emission sector were calculated compared to the sum of all five emissions sector contributions."

*Technical comments:*
*Please recheck super- and sub-script for air pollutants.*

Thanks for this comment. We revised the manuscript accordingly.